

# Source regions contributing to excess reactive nitrogen deposition in the Greater Yellowstone Area (GYA) of the United States

Rui Zhang[1], Tammy M. Thompson[2], Michael G. Barna[3], Jennifer L. Hand[1], Jill A. McMurray[4], Michael D. Bell[3], William C. Malm[1], Bret A. Schichtel[3]

[1]Cooperative Institute for Research in the Atmosphere, Colorado State University, Fort Collins, CO, 80521, USA

[2]American Association for the Advancement of Science, Washington DC, 20005, USA

[3]Air Resource Division, National Park Service, Denver, CO, 80225, USA

[4]US Forest Service, Bozeman, MT, 59771, USA

*Correspondence to*: Bret A. Schichtel (bret_schichtel@nps.gov)

**Abstract.**

Research has shown that excess reactive nitrogen ($N_r$) deposition in the Greater Yellowstone Area (GYA) of the United States has passed critical load thresholds and is adversely affecting sensitive ecosystems in this area. To better understand the sources causing excess $N_r$ deposition, the Comprehensive Air Quality Model with extensions (CAMx), using Western Air Quality Study (WAQS) emission and meteorology inputs, was used to simulate $N_r$ deposition in the GYA. CAMx's Particulate Source Apportionment Technology (PSAT) was employed to estimate contributions from agriculture (AG), oil and gas (OG), fire (Fire), and other (Other) source sectors from 27 regions, including the model boundary conditions (BC) representative of international contributions, to the simulated $N_r$ for 2011. Emissions from the AG and Other source sectors are predominantly from reduced N and oxidized N compounds, respectively. The model evaluation revealed a systematic underestimation in ammonia ($NH_3$) concentrations by 65% and overestimation in nitric acid concentrations by 108%. The measured inorganic N wet deposition at National Trend Network sites in the GYA was overestimated by 31–49%, due at least partially to an overestimation of precipitation. Source apportionment results showed that the AG sector was the single largest contributor to the GYA total $N_r$ deposition, contributing 34% on an annual basis. Seventy-four percent of the AG contributions originated from the Idaho Snake River valley, with Wyoming, California, and northern Utah contributing another 7%, 5%, and 4% respectively. Contributions from the OG sector were small at about 1% over the GYA, except in the southern Wind River Mountain Range during winter where they accounted for more than 10%, with 46% of these contributions coming from OG activities in Wyoming. Wild and prescribed fires contributed 18% of the total $N_r$ deposition, with fires within the GYA having the highest impact. The five largest source area contributions to the annual total $N_r$ deposition in the GYA were 1) the Snake River valley (38% with AG 68%, OG 2%, Fire 15%, and Other 16%); 2) BC (21%); 3) Wyoming (12% with AG 19%, OG 5%, Fire 38%, and Other 39%); 4) California (7% with AG 26%, OG 1%, Fire 14%, and Other 59%); and 5) northern Utah (6% with AG 25%, OG 2%, Fire 10%, and Other 63%). These results suggest that $N_r$ deposition over the GYA, especially in the western region, was above the critical loads for sensitive ecosystems, and



AG from the Snake River valley was the largest contributor. Distant source regions were also important, with large contributions from the BC, i.e., international source regions.

## 1 Introduction

The Greater Yellowstone Area (GYA) (see Figure 1) of the United States, with Yellowstone National Park (YNP) and Grand Teton National Park (GTNP) at its core, is one of the largest remaining intact ecosystems in the northern temperate zone and features diverse wildlife, alpine lakes, forests, and geologic wonders (Keiter and Boyce, 1994; NPS, 2017). Increasing concentrations of reactive nitrogen ($N_r$) compounds in air, rain, and snowpack samples over the GYA have been reported in the past 30 years and linked to $N_r$ emissions from human activities (Clow et al., 2003; Blett et al., 2011; IMPROVE, 2011; Sullivan et al., 2011; USGS, 2014; NADP, 2016; Nanus et al., 2017; also, see Figure S1). The inorganic wet $N_r$ deposition rates measured at high-elevation National Trend Network (NTN) sites within the GYA in 2010 were 2.5–3.5 kg N ha$^{-1}$ yr$^{-1}$, compared with 1.5–2.5 kg N ha$^{-1}$ yr$^{-1}$ in 2000 (NADP, 2016). This is relevant to the long-term conservation of the area because as $N_r$ deposition levels increase, they can cross critical load thresholds, at which negative effects to sensitive ecosystem components can occur (Porter et al., 2005; Pardo et al., 2011). Additional concerns posed by enhanced $N_r$ deposition include lake acidification, loss of lichen biodiversity, and eutrophication (Baron, 2006; Blett et al., 2011; NADP, 2016). While ecosystem changes due to excess $N_r$ deposition over Class I areas including the GYA have been documented (e.g., Baron et al., 2011; Saros et al., 2011; Sullivan et al., 2011; Spaulding et al., 2015; Nanus et al., 2017), the origins, chemical composition, and spatial and temporal changes in the deposition over this region are not as well understood.

Total $N_r$ is a mix of oxidized and reduced inorganic nitrogen (N) and organic N compounds that are chemically and biologically active in the Earth's biosphere and atmosphere and are deposited through wet and dry processes. These compounds arise from a variety of sources, with inorganic oxidized N primarily emitted as nitrogen oxides ($NO_x$) from fossil fuel combustion, with 25% from power plants, 50% from automobiles, and 10% from other mobile sources (EPA, 2015). Atmospheric reactions of $NO_x$ result in nitric acid ($HNO_3$), particulate nitrate, and other compounds. Reduced N arises primarily from ammonia ($NH_3$) gas emissions from agricultural activities, which can react with acidic aerosols to form ammonium ($NH_4^+$) compounds (Galloway et al., 2004). Mobile sources are also an important source of $NH_3$ and can be the primary emitter in urban areas. A recent study found the increasing importance of on-road emissions of $NH_3$, which at 40% exceed agricultural emissions (Fenn et al., 2018). There are hundreds of organic N compounds, including reduced (e.g., amines) and oxidized (e.g., alkyl nitrates) forms. Sources of organic N are less well known, but increasing evidence suggests that biomass burning and agriculture are significant contributors, as are atmospheric reactions of $NO_x$ with volatile organic compounds (Cape et al., 2011; Reay et al., 2012). With the steady decline of $NO_x$ emissions in the United States during past decades as a result of the implementation of the Clean Air Act, the importance of reduced N to the total N



deposition budget has increased (Li et al., 2016). Specific to the GYA, local anthropogenic emissions are small, but upwind sources, including agricultural activities in the Snake River valley and northern Utah, wildfires throughout the western United States, energy development in the Upper Green River Basin, and anthropogenic activities at urban centers such as Salt Lake City, are larger and likely to be significant contributors to regional N emissions (Prenni et al., 2014).

To better understand the levels and composition of the $N_r$ compounds deposited in the GYA and to help guide strategies to reduce N deposition, the National Park Service (NPS) initiated the Grand Teton Reactive Nitrogen Deposition Study (GrandTReNDS), which included spatially and temporally detailed measurements of N compounds during April to September 2011 (Benedict et al., 2013a; Prenni et al., 2014). It was found that during summer months at the high-elevation sites (e.g., Grand Targhee; see Figure 1), 62% of the nitrogen deposition was from reduced N and about equally split between dry and wet deposition, and oxidized N only accounted for 27% of the nitrogen deposition budget, with the remaining in the form of wet-deposited, organic N. Study findings indicate a significant west-to-east gradient in atmospheric $NH_3$ concentrations, with higher concentrations west of the Teton mountain range. Concurrently measured concentrations of $HNO_3$ and $PM_{2.5}$ (particulate matter with aerodynamic diameter less than 2.5 μm) nitrate, and $NH_4^+$ showed relatively small west-to-east gradients inside GTNP (Benedict et al., 2013a; Prenni et al., 2014).

The origins of $N_r$ transported to the GYA and other remote locations in the western United States have been examined in past modeling studies. Back trajectory analyses have shown that airmass transport to GTNP is predominantly from the west through the Snake River valley and from the southwest through northern Utah (Prenni et al., 2014). Zhang et al. (2012) applied the global Chemical Transport Model (CTM) GEOS-Chem (Bey et al., 2001) for zero-out sensitivity simulations and found that in 2006 natural sources, including lightning and wildfires, contributed more than 10% of the total $N_r$ deposition over the Teton area. Lee et al. (2016) used the adjoint version of GEOS-Chem to quantify the sources of $N_r$ deposition in eight selected federal Class I areas in 2010, including GTNP, and found the importance of emissions from California to $N_r$ deposition in remote areas in the western United States. Mobile $NO_x$ and livestock $NH_3$ were also found to be major sources of $N_r$ deposition in GTNP.

Rocky Mountain National Park (RMNP), located in the Intermountain West, has been the focus of several Nr deposition studies. This area has many similarities to the GYA, although the intense agricultural activity and large population centers are located to the east of RMNP as opposed to the west as is the case for the GYA. In one 2009 modeling study, Thompson et al. (2015) found that 40% of the reduced N deposition originated from ammonia sources within Colorado. The emissions from more-distant sources in California and the Snake River valley in Idaho as well as international sources each contributed 7–15% of the total $N_r$ deposition. In a hybrid modeling technique, Malm et al. (2016) combined the source attribution results from Thompson et al. (2015) with measured $N_r$ concentrations and found that $N_r$ contributions to RMNP were also predominantly from the sources within Colorado, with a significant portion (27%) originating from sources along the Front Range of Colorado. Furthermore, they pointed out that reduced $N_r$ constituted 66% of the total deposition budget.



In this work, we add to the growing body of $N_r$ modeling source apportionment studies by conducting a detailed analysis using the Particulate Source Apportionment Technology (PSAT) module within the CAMx (Comprehensive Air Quality Model with extensions) (Ramboll Environ, 2014) CTM to quantify the seasonal contributions from different source regions and source sectors to $N_r$ throughout the GYA. The model simulation of $N_r$ and its constituents were first evaluated against routine measured data as well as the unique data measured during the GrandTReNDS campaign period (Benedict et al., 2013a; Prenni et al.,

2014). The final source apportionment results are then interpreted within the context of the identified model bias and uncertainties.

## 2 Modeling system for $N_r$ source apportionment

Modeling simulations for 2011 were conducted using the CAMx version 6.10 (ENVIRON, 2014) with two nested grids. The outer domain (36 km) covered the contiguous United States (CONUS), as well as portions of Canada and Mexico, while the inner domain (12 km) encompassed the western United States and focused on states within the Western Regional Air Partnership (WRAP) (see Figure 1).

The hourly meteorological inputs for 2011 were generated by the Weather Research and Forecasting (WRF) model (WRF-ARW, version 3.5.1) (Skamarock et al., 2008) and were obtained from the Intermountain West Data Warehouse (IWDW) (http://views.cira.colostate.edu/tsdw/). This meteorological simulation performed comparably to other recent prognostic model applications used in air quality planning (UNC-Chapel Hill and ENVIRON, 2014a).

The emission inventory used by CAMx was primarily derived from 2011 National Emissions Inventory version 2 (NEI2011v2) (EPA, 2015) with the Sparse Matrix Operator Kernel Emissions (SMOKE) processing system version 3.0 (Houyoux et al., 2002) for anthropogenic emissions, the Model of Emissions

of Gases and Aerosols from Nature (MEGAN) version 2.10 (Guenther et al., 2012) for biogenic emissions, and the WRAP Windblown Dust Model (WRAP-WBD) to estimate wind-driven dust emissions (UNC-Chapel Hill and ENVIRON, 2014b). Emissions from the oil and gas sector were further updated by the IWDW to represent the best available inventory for oil and gas activity in the western United States at the time of modeling (UNC-Chapel Hill and ENVIRON, 2014b). The emissions for fire activities include agricultural fires, prescribed fires, and wildfires and were generated by the Particulate Matter Deterministic and Empirical Tagging and Assessment of Impacts on Levels (PMDETAIL) study (Moore et al., 2012). PMDETAIL developed 2011 fire emissions using satellite

data, ground detects, and burn scar and estimated the plume rise depending on fire size and type (Mavko and Morris, 2013).

The boundary conditions for the 36-km domain were estimated from a 2011 global model run using the Model for Ozone and Related chemical Tracers (MOZART) version 4.6 (Emmons et al., 2010). The simulation year of 2011 was preceded by 15 days of "spin-up" time to minimize the effects of initial



conditions. A more detailed description of the WRF-SMOKE-CAMx modeling platform applied in this study is summarized in Table S1 as well as the 2011 Three-State Air Quality Study (3SAQS) (UNC-Chapel Hill and ENVIRON, 2014b).

For the source apportionment estimates, 27 source regions, as well as the lateral boundary conditions (BC), were "tagged" in the CAMx PSAT simulation. Figure 1 provides the source region partition map of the inner 12-km domain emissions. In addition, the emissions for each region were further subdivided into four source sectors of 1) agriculture (AG), 2) oil and gas activity (OG), 4) fire activity including wildfires and prescribed fires (Fire) and 4) the remaining sources labeled as Other. The Other source sector primarily comes from mobile and large point sources with smaller contributions from natural sources such as lightning. Table S2 provides the estimated annual $NH_3$ and $NO_x$ emissions in the corresponding source regions and sectors. The regions were selected to highlight important source sectors contributions to $N_r$ deposition in the GYA. For example, the state of Wyoming (WY) was partitioned into five regions (YNP, Jackson, Upper Green River, and Eastern and Western WY) to differentiate the possible source impacts from urban activity in Jackson as compared to energy development in southwestern Wyoming (Blett et al., 2011; NPS, 2017). Significant agricultural operations in the Snake River valley in Idaho, northern Utah, and northeastern Colorado were tagged due to their high ammonia emissions (see Figure 1) associated with fertilizer application and confined animal feeding operations (Fenn et al., 2003; Clarisse et al., 2009; Prenni et al., 2014). Lastly, wildfires are episodic events (http://wrapfets.org/map.cfm) that can have large intermittent contributions to $N_r$ deposition, but they can mask important contributions from other sources that are significant in nonfire years.

CAMx-PSAT treats nitrogen-containing compounds as one of seven species: gaseous $NH_3$; particulate ammonium ($PNH_4$); reactive gaseous nitrogen (RGN), which includes primary emissions of $NO_x$, nitrous acid (HONO), nitrate radical ($NO_3$), and dinitrogen pentoxide ($N_2O_5$); gaseous nitric acid ($HNO_3$); gaseous peroxy nitrogen (TPN), including peroxyacetyl nitrate (PAN) and peroxynitric acid (PNA); gas-phase organic nitrate (NTR); and particulate nitrate ($PNO_3$). PSAT maintains the source group identity (i.e., source region and source sector) by apportioning the secondary species to the precursor emissions (ENVIRON, 2014). In the source apportionment comparison results, we report the reduced $N_r$ deposition as the sum of $NH_3$ and $PNH_4$ and the oxidized $N_r$ deposition as the sum of RGN, $HNO_3$, $PNO_3$, TPN, and NTR in units of kg N ha$^{-1}$.

## 3 Evaluation of CAMx-simulated $N_r$ concentration and deposition rates

Acceptable model performance of the regional air quality modeling system is a prerequisite for a credible source apportionment interpretation (Boylan and Russell, 2006; EPA, 2014; Emery et al., 2017). In this work, the CAMx simulation was extensively evaluated against routine monitoring data as well as data collected in the GrandTReNDS special field study (Benedict et al., 2013a; Prenni et al., 2014) and against the nitrogen deposition estimates from the NADP total



deposition maps (TDEP) hybrid modeling results (Schwede and Lear, 2014). The performance metrics recommended from the EPA's modeling guidance for ozone, $PM_{2.5}$, and regional haze attainment demonstrations (Yu et al., 2006; EPA, 2014) were used (see Table 1).

The variables and routine monitoring networks used in the model evaluation were $NH_3$ concentrations from the Ammonia Monitoring Network (AMoN) (http://nadp.sws.uiuc.edu/AMoN/); nitric acid ($HNO_3$), $PNO_3$, and $PNH_4$ concentrations as well as estimated dry deposition fluxes from the Clean Air Status and

Trends Network (CASTNet) (https://www.epa.gov/castnet); $PNO_3$ and $PNH_4$ concentrations from the Chemical Speciation Network (CSN) (https://www3.epa.gov/ttnamti1/speciepg.html); $PNO_3$ concentrations from the Interagency Monitoring of Protected Visual Environments (IMPROVE) network; and wet-deposited inorganic oxidized ($NO_3^-$) and reduced ($NH_4^+$) nitrogen and associated precipitation rates at National Atmospheric Deposition Program (NADP) NTN sites. Each network had a unique sampling frequency and duration (Table 1). The hourly CAMx outputs were aggregated to match the timescales of the measured data. All data flagged as questionable were removed from the analysis. In general, the Clean Air Status and Trends Network (CASTNet),

IMPROVE, AMoN, and NADP networks sample in rural areas, while the data from the CSN network primarily represent the air quality in urban and suburban settings. Although organic N species were also measured in the GrandTReNDS campaign, we focus on the inorganic N budget comparison, given the large uncertainties for organic N prediction (Jickells et al., 2013) and its incomplete treatment in the model's chemical mechanism. For example, the modeling system does not account for primary emissions of organic N compounds but does include the formation of organic N from the alkylperoxy radical and secondary alkoxy radical (ENVIRON, 2014).

**3.1 Evaluation against data in the GYA**

3SAQS performed photochemical grid modeling using the same modeling platform and input files as this study (UNC and ENVIRON, 2014b) and evaluated the model performance over the western United States. A subset of these results is presented in Supplement File S1. Model performance statistics for the nitrogen species within the GYA area at AMoN, CASTNet, IMPROVE, and NTN network sites (Figure 1) at different periods in 2011 are presented in Table 1. The biases at the GYA sites are similar to those throughout the West (Table in File S1) in that the CAMx simulation significantly overestimated the $HNO_3$ with NMB

= 108% and significantly underestimated the $NH_3$ concentrations with NMB = -65%. While the model had skill in reproducing the daily variation in $HNO_3$ with correlation coefficient $r = 0.71$, it had little skill for $NH_3$ with $r = 0.2$. The overestimation of $HNO_3$ has also been reported in other modeling studies (e.g., Zhang et al., 2012; Lee et al., 2016) and may be due to excessive $N_2O_5$ hydrolysis in the model (Heald et al., 2012). The poor $NH_3$ results may realted with the high





uncertainty in the NH$_3$ emission inventory (Clarisse et al., 2009) and missing important physical mechanisms in the model, including the lack of bidirectional NH$_3$ deposition (Zhang et al., 2010; Bash et al., 2013; Thompson et al., 2015; Zhu et al., 2015).

For PNO$_3$ and PNH$_4$ simulations in the GYA, the CAMx underestimated both species, with better performance for PNH$_4$ than PNO$_3$ (3% versus 37%, respectively, in terms of normalized mean bias (NMB)) and better agreement for PNO$_3$ at CASTNet sites versus IMPROVE sites (37% versus 58% for NMB, respectively). The errors and biases in the dry deposition fluxes compared to CASTNet values follow the same patterns as in the ambient concentrations, but it should be noted that CASTNet and CAMx use different algorithms to estimate dry deposition velocities, and these model-to-model discrepancies will manifest themselves in the performance evaluations. Overall, the CAMx model performance for the concentrations of gaseous and PM nitrogen components in the GYA fall within the range of reported regional air quality model performance metrics from other peer-reviewed studies (Simon et al., 2012).

Wet deposition measurement from the five NTN sites with sufficient data was available from within the GYA. Comparisons to CAMx showed that the model captured the general trends in these data with $r \sim 0.32$–0.34 and but were somewhat biased with NMB = 31% for NO$_3^-$ and NMB = 49% for NH$_4^+$. The precipitation simulations were consistently 100–200% higher than the rain gauge measurements at the NTN sites, showing that WRF overestimated the frequency and intensity of precipitation events over the GYA in 2011. However, note that 2011 was a large snowpack year; by May, much of the GYA was sitting at 100–180% of normal snow weather equivalent (USGS, 2014). Precipitation measurements tend to be low during high snow events.

The seasonal, simulated ambient concentrations and deposition rates are compared against measured CASTNet and NADP data at the YNP and Pinedale monitoring sites in Figure 2. The significant overestimation in HNO$_3$ is evident in all seasons. Also evident is the poor simulation of the seasonality in N$_r$ deposition, primarily due to the poor reproduction of the wet deposition, which is at least partly due to the large errors in the simulated precipitation.

### 3.2 Evaluation against GrandTReNDS data

The GrandTReNDS campaign provides a unique opportunity to evaluate the capability of CAMx to simulate the N$_r$ compounds and deposition budget. Detailed measurements, including NH$_3$, were made at three sites that crossed GTNP from west to east: Driggs, in the foothills just west of GTNP (43.74 °N, -111.87 °W, elevation 1947m); Grand Targhee, an upper elevation site on the western edge of GTNP (43.78 °N, -110.94 °W, elevation 2722m); and the National Oceanic and Atmospheric Administration (NOAA) Climate Station site on the eastern edge of GTNP (43.66 °N, -110.71 °W, elevation 1978m) (also see Figure 1). Figure 3 presents the monthly deposition budgets for these three sites during the sampling periods. As shown, the simulation does a poor job of reproducing the total N$_r$ deposition rates both in the month-to-month variation as well as across the sites. However, consistent with the observations, the simulation shows that wet deposition is larger than dry and that the contribution from reduced N deposition was larger than from the oxidized N deposition at all three sites, although the



observed range of 70–80% reduced N was more than the 55–68% simulated in CAMx. The primary cause of this bias was the overestimation in the $HNO_3$ dry deposition rates, which were 2–3 times larger than those derived from the measured data. This is consistent with the systematic overestimation of $HNO_3$ concentrations (Table 1). Other biases also exist, including an underestimation in the $NH_3$ dry deposition, which was somewhat balanced by an overestimation in the $NH_4^+$ wet deposition.

An additional challenge that affected model performance was the difficulty in estimating precipitation rates. This is shown in Figure 3, where the simulated precipitation rates do not reproduce the month-to-month variation and generally were highly overestimated. For example, on average the simulated precipitation at Driggs was more than double the measured precipitation, and it was more than a factor of 4 higher at the NOAA Climate Station site.

### 3.3 Evaluation against NADP-TDEP

TDEP maps (Schwede and Lear, 2014) are widely used in the land management community to assess total $N_r$ deposition throughout the United States and
estimate the critical load exceedances in sensitive ecosystems (Saros et al., 2011; Nanus et al., 2017). TDEP employs a hybrid approach to integrate measurements from multiple networks, including CASTNet and NTN, with Community Multiscale Air Quality (CMAQ) modeling (Byun and Schere, 2006) results for deposition velocities and unmeasured species' dry deposition, as well as PRISM (Parameter-elevation Relationships on Independent Slopes Model) (Daly et al., 1994) for high-resolution precipitation estimates to mapping total deposition in the United States (Schwede and Lear, 2014). Both the CAMx simulation in this study and the TDEP results are derived from model simulations and subject to similar errors in emissions and physical and chemical processes.
However, with the incorporation of measured wet $N_r$ deposition and N concentration data into the TDEP results, they are expected to be less biased than the deposition results from pure CAMx simulation.

        The TDEP total $N_r$ deposition and the CAMx 2011 simulation in this work exhibited similar spatial and temporal patterns across the western United States; for example, both sets of results show high $N_r$ deposition in the Snake River valley, northern Utah, and across the Wyoming state border area near GTNP with values >5 kg N ha$^{-1}$yr$^{-1}$. Within the GYA (Figure S4), the CAMx simulation had higher dry $N_r$ deposition, which was more spatially heterogeneous than the
corresponding TDEP results, with significantly higher $N_r$ deposition in the agricultural lands to the west of the GYA and hot spots due to wildfires that are not evident in the TDEP results. Both sets of results showed higher $N_r$ wet deposition at the higher-elevation sites in the interior of the GYA, which was associated with higher precipitation rates. However, the TDEP $N_r$ wet deposition was generally higher throughout the GYA, with an annual average $N_r$ wet deposition rate of 2.0 N ha$^{-1}$yr$^{-1}$ versus 1.3 N ha$^{-1}$yr$^{-1}$ from CAMx. Both precipitation maps across the GYA generated by WRF and PRISM had similar spatial patterns, with hotspots located in high-elevation mountain ranges, though the WRF annual precipitation rates were on average 73% higher than the PRISM estimates.



The annual Nr deposition budget and the annual precipitation rate from TDEP and the CAMx simulations at eight Class I areas over the GYA area are compared in Figure 4. The reported CAMx dry and wet Nr deposition values at the eight Class I areas in Figure 4 are the average of the simulation values at corresponding grid cells for each area. Generally, results from the CAMx model agreed well with TDEP results in terms of replicating the spatial gradients and the ratios of oxidized versus reduced N deposition. The TDEP 2011 annual $N_r$ deposition at the GYA receptor sites was in the range of 2.8–5.4 kg N ha$^{-1}$yr$^{-1}$, while the corresponding values for CAMx were 2.2–4.3 kg N ha$^{-1}$yr$^{-1}$. Both results showed the west-to-east gradient (Prenni et al., 2014) with higher $N_r$ deposition at the western side of the GYA and relatively low values at Fitzpatrick Wilderness. Also, both models showed the importance of reduced $N_r$ in the GYA with a nearly 50% or higher contribution to the total $N_r$ deposition budget. However, the two models differed on the ratio of dry versus wet $N_r$ deposition, with CAMx simulating a higher fraction from dry $N_r$ deposition than TDEP.

## 4. Source apportionment of $N_r$ deposition over the GYA in 2011

The seasonal modeled $N_r$ deposition budgets averaged over the GYA are presented in Figure 5. As shown, the total $N_r$ deposition rates peaked in the summer (1.12 kg N ha$^{-1}$season$^{-1}$) with somewhat lower rates in the spring(0.91 kg N ha$^{-1}$ season$^{-1}$) and fall (0.81 kg N ha$^{-1}$ season$^{-1}$) and with winter rates (0.29 kg N ha$^{-1}$ season$^{-1}$) being about a factor of 3 smaller than in the other seasons. These patterns are similar to the measured and modeled data presented in Figure 2. In total, the annual model $N_r$ deposition was 3.13 kg N ha$^{-1}$yr$^{-1}$, with wet deposition accounting for only ~40%. Reduced N compounds were the largest contributor, except in winter, which is consistent with past studies (Li et al., 2017). Contributions from organic N compounds are not measured in routine monitoring programs. Together they accounted for <10% of the $N_r$ deposition, suggesting a small but not insignificant contribution. This is also less than has been measured in field studies conducted at GTNP (Benedict et al., 2013a; Prenni et al., 2014) and in RMNP (Benedict et al., 2013b), where the GrandTReNDS study showed on average 8–18% contribution from organic N to total $N_r$ deposition budgets during the whole campaign period and up to 39% in June at the NOAA Climate Station site (Figure 7 in Benedict et al., 2013a).

The relative contributions from the four modeled source sectors (AG, OG, Fire, and Other) and the BC averaged over the GYA are presented in Figure 6, while Figure 7 presents the seasonal and spatial pattern of their contributions over the GYA. As shown in Table S2, the AG source sector was composed of almost all reduced N compounds (>99%), while the Other source sector was primarily composed(97%) of oxidized nitrogen compounds, with about 88% originating from anthropogenic combustion emissions, including point and mobile sources, and the remainder from the natural emissions from soil and lightning. Contributions from the Fire and the BC were more evenly split between reduced and oxidized N contributions.





Reduced N from the AG source sector was the largest contributor in spring (40%) and fall (41%) seasons, while oxidized N from the Other source sector was the largest contributor in summer (29%) and winter (44%) (Figure 6). In terms of geographic impact (Figure 7), AG emissions contributed as much as 80% of the total $N_r$ deposition in the western portion of the GYA during the spring and fall, which was associated with the outflow from the Snake River valley. In the model, $NH_3$ from regional agriculture activities was treated as from surface area sources (i.e., emitted into the first model layer, which is approximately 24 m

thick).  These low-level emissions can be quickly deposited to the surface unless there is sufficient vertical mixing to inject the $NH_3$ into the upper levels of the atmosphere (Ferm, 1998; Fenn et al., 2003) or if it reacted with acidic gases and aerosols.  Consequently, it is likely that a higher fraction of the modeled $NH_3$ emissions from AG sector will be deposited in the lower-elevation periphery of the GYA near the agricultural lands and not impact the more-distant mountainous interior (Figure 1). The incorporation of the bidirectional $NH_3$ flux could extend the $NH_3$ emission footprint (Bash et al., 2013; Zhu et al., 2015).

The OG source sector contributed only about 1% of the total $N_r$ deposition over the GYA, with contributions of 10% or more occurring during winter in

the southeastern corner of the GYA where nearby OG activity in the Jonah Field and Pinedale Anticline was taking place. Wildfires are episodic and their locations and magnitudes vary significantly from year to year (Westerling and Swetnam, 2003; Parisien et al., 2012). In 2011, fire events contributed on average 18% of the total $N_r$ deposition in the GYA. Most of the wildfire happened in summer and fall, while agriculture and prescribed burning occurred in winter and spring. Near the fire activities, the contribution to the $N_r$ deposition could be more than 90%, as seen in Figure 7.  The Other source sector had relatively uniform contributions throughout the GYA, indicative of contributions from regional sources.  The Other sector accounted for 26% of the annual $N_r$ deposition, with its

largest absolute contributions in the summer, but the highest relative contribution in the winter at 44% when AG contributions were at their lowest. Finally, the BC had high contributions, often over 20%, with the highest contributions occurring in the northern part of the GYA and at higher-elevation sites.

The seasonal contributions from the modeled source regions and sectors to the average total $N_r$ deposition over the GYA are summarized in Figure 8. As shown, the Snake River valley in Idaho was the largest contributor (in all seasons), with annual mean contributions of 38% and a maximum contribution of 43% in fall. Most (74%) of the $N_r$ from this region was from the AG source sector and was composed of reduced N (Table S4). The next four largest contributors, on

average, were the BC (21%), western Wyoming (8%), California (7%), and northern Utah (6%). The impact of emissions from Wyoming to the GYA during summer and fall (14% and 16%, respectively) was more pronounced than winter and spring (5% and 7%, respectively). The contributions of long-range transport from California and the BC were higher during spring and winter.

Seasonal source apportionment results of the average dry and wet $N_r$ deposition over the GYA are shown in Figures 6 and 8. Compared to the results for total $N_r$ deposition, the dry $N_r$ deposition had higher contributions from closer sources, such as the Snake River valley (46% for dry versus 38% for total), with

emissions primarily from AG sources. Similarly, contributions to dry $N_r$ deposition from Wyoming were 15% compared to 12% for total $N_r$ deposition and



ranked as the second largest contributor. The contributions from distant source regions decreased.  For example, the BC decreased from 21% for total $N_r$ deposition to 12% for dry $N_r$ deposition.

The opposite pattern is seen for wet $N_r$ deposition, where the contributions from the distant source regions increased relative to the neighboring ones. The annual contributions from the BC increased to 34% and peaked in spring and summer at 37%, associated with higher precipitation amounts than the other two seasons. Annual contributions from sources in California (10%) and Utah (8%) surpassed Wyoming (7%). Furthermore, the seasonal variation for wet Nr deposition was different from dry and total $N_r$ deposition, with the highest deposition rates occurring in spring as opposed to summer.

The GYA has been the focus of several ecological assessments of the response of ecosystems to changing $N_r$ deposition levels (Spaulding et al., 2015; Nanus et al., 2017). Figure 9 presents the source attribution results for ten sites within the GYA where either ecosystem response studies or deposition monitoring have been conducted for lichen diversity, alpine lake chemistry, and snow pack analysis. In Table 2, the critical load (CL) values are provided as a range of lower end and upper-end estimates of the annual total inorganic $N_r$ deposition values (Lynch et al., 2015) with confidence levels (Pardo et al., 2011;).  The simulated $N_r$ deposition exceeded the lower CL values at 3 of the 10 sites, specifically, Tower Falls, Holly Lake, and Pinedale. Comparatively, the 2011 TDEP $N_r$ deposition results exceeded the CL in 6 out 10 sites (sites at Pinedale, Holly Lake Twin Island, Biscuit Basin, Jedediah Smith Wilderness, and Black Joe Lake). As shown in Figure 9, the sites that exceeded the CL tend to be in high alpine locations with four of these sites on the western slope of the mountains which are downwind of the Snake River valley.  These results are consistent with another modeling study to access CL exceedances in Class I areas using GEOS-Chem (Ellis et al. 2013; Lee et al., 2016).  In addition, in one study (Nanus et al., 2017) over 30% of the GYA was estimated to potentially exceed lower Nr deposition CL thresholds, with the greatest impacts in sensitive high elevation basins including areas within National Parks and Wilderness.

In terms of emission sectors and source regions contributing to the total annual $N_r$ deposition at CL exceedance sites, emission sources from the Snake River valley were the largest contributor (27–32%) and AG emissions were the largest source of this subset. The next three largest contributors were transport from the BC (23–25%) and emissions from northern Utah (8–15%) and California (7–8%). Wyoming emissions associated with the OG and Fire emission sectors contributed around 3–5% and 14–23%, respectively, of the $N_r$ budget for receptor sites at the southeastern corner of the GYA.

**5. The influence of model bias on source apportionment results**

It is evident from the results in section 4 that the attribution of total Nr deposition to source the source regions and sectors are sensitive to $NH_3$ dry deposition rates; the relative contributions of dry and wet deposition; and the concentrations of N compounds from BC. However, the model evaluation revealed a significant underestimation of $NH_3$ concentrations and overestimation of $HNO_3$ concentrations and precipitation rates; thus these modeling errors could bias the



source attribution results. To better understand the potential effects of these biases, sensitivity analyses of the source attributions to changes in NH$_3$ dry deposition rates and average precipitation rates as well as potential biases in the BC were evaluated.

To test the sensitivity of the apportionment to NH$_3$ dry deposition rates, the deposition velocities were reduced by increasing the NH$_3$ resistance scaling factor by 10%, following the methodology used in Thompson et al. (2015). The Zhang et al. (2003) dry deposition scheme was used in the CAMx simulations (Table S1), and this resistance scaling factor is designed to address the rapid removal of "sticky" compounds such as HNO$_3$ and NH$_3$ and can yield a nonlinear response in the estimated dry deposition velocity. July and August 2011 were simulated using the modified deposition velocity, and these results will be referred to as "DV_0.1". The 10% change in the resistance factor slowed the NH$_3$ deposition velocity from 2.5~4 cm s$^{-1}$ to 1~1.5 cm s$^{-1}$ over the GYA, resulting in values more comparable to those used in the GrandTreNDS study (Benedict et al., 2013a; Prenni et al., 2014). The simulated NH$_3$ concentrations for the DV_0.1 case increased throughout the GYA compared to the base case. This resulted in a better agreement with NH$_3$ measurements at the Grand Targhee and NOAA Climate Station sites but poorer agreement at the Driggs monitoring site (Figure S5). The slower dry deposition velocities result in a longer NH$_3$ lifetime, allowing it to travel farther from nearby source regions, e.g., the Snake River valley, into the GYA, and cause a more homogeneous concentration pattern throughout the GYA (Figure S6). As shown in Figure 10, the slower deposition velocities also somewhat altered the source attribution results. The contribution from the AG emission sector increased with the DV_0.1 simulation to 23% compared to 19% in the base case, with a smaller decrease in the contribution from the Other and the Fire sector. This change was due to small increases in the contributions from the Snake River valley and northern Utah and decreases from Wyoming. Overall, decreasing the NH$_3$ dry deposition rate by about a factor of 2 had only a small impact on the N$_r$ deposition budget and source apportionments results in the GYA. It is important to note that, although this was a significant reduction in the simulated dry deposition velocity for NH$_3$, it still represents a relatively rapid removal rate as compared to other species, and NH$_3$ is quickly lost from the atmosphere in either case. It is known that NH$_3$ deposition in many environments is a bidirectional as opposed to a unidirectional process, and modeling the NH$_3$ flux as a bidirectional process may further decrease the bias for ambient NH$_x$ (NH$_x$=NH$_3$+PNH$_4$) concentration simulations (Bash et al., 2013; Wen et al., 2014; Whaley et al., 2018). The key process in air quality models to represent the re-emission of NH$_3$ from soil and plants to the atmosphere is the estimation of the available soil NH$_x$ pool and the parameterization of compensation points for the conditions to re-emit NH$_3$ (Zhang et al., 2010; Whaley et al., 2018). In the CMAQ model, the bidirectional NH$_3$ deposition was realized by coupling with the United States Department of Agriculture's (USDA) Environmental Policy Integrated Climate (EPIC) agroecosystem model to provide the fertilization timing, rate, and composition (Bash et al., 2013). There is no similar parameterization available in the current CAMx model. Furthermore, the CAMx source apportionment tools cannot properly account for the origin of NH$_3$ concentrations at a receptor that has been deposited then re-emitted.



The CAMx simulation overestimated the wet $N_r$ deposition at measured sites, which was likely associated with an overestimation in the precipitation rates from WRF, especially at high elevation sites. This precipitation rate bias was large, with the annual precipitation over the GYA more than 73% higher than the PRISM estimates. We used the Noah land surface model and Kain-Fritsch scheme cumulus parametrization in the WRF simulations (Table S1), and those physical module configurations were reported to have the tendency to overestimate precipitation (Warrach-Sagi et al., 2013). To evaluate the impact of the

overestimation in precipitation on the source attribution results, the seasonal wet deposition rates were scaled to the measured precipitation rates at all NADP-NTN and GrandTReNDS monitoring sites, following the procedures by Appel et al. (2011). This was equivalent to scaling the modeled wet deposition rates by the ratio of the measured to modeled precipitation rates. This approach assumes that the concentrations of $N_r$ in the precipitation were the same in the model and measured data, which was not the case. After the precipitation adjustment, the correlation between the simulated and measured $N_r$ wet deposition improved (Figure S7). Within the GYA, however, the scaled $N_r$ wet deposition underestimated the measured by about a factor of 2 and significantly underestimated the

ratio of wet to dry deposition. Consequently, scaled wet deposition results were not used in this assessment.

The BC used in this work was derived from a MOZART global model simulation. An alternative set of BC from the GEOS-Chem global model was also evaluated. Both sets of BC resulted in high contributions to the total $N_r$ deposition in the GYA, with the GEOS-Chem results having a slightly higher average contribution of 23% compared to 21% for MOZART (Figure S8). However, the GEOS-Chem BCs resulted in higher relative contributions of oxidized N to the total Nr deposition rate compared to the MOZART BCs (51% and 45%, respectively). The poor correspondence in the oxidized to reduced $N_r$ split is

reflective of the large uncertainties in the BC contributions to the $N_r$ deposition and suggests that more evaluation of the global model results is warranted.

To examine the potential bias in the BC contributions, the simulated $PNO_3$ concentrations were compared to measurements from the IMPROVE monitoring program over the western United States for 2011. This comparison is shown in Figure 11, where the ratio of the simulated to measured $PNO_3$, i.e., an estimate of the bias, is plotted against the relative fraction of the contribution of the BC to the simulated $PNO_3$. The data were first segregated by the fractional contribution of the BC and then averaged together. As shown, for the MOZART BC, the bias increased with larger relative contributions from the BC, and when

the BC fraction is 60%, the bias was more than a factor of 2. This suggests that at least the particulate nitrate concentrations from the BC are overestimated and possibly other $N_r$ compounds from the BC as well. In a CMAQ simulation using BC derived from a GEOS-Chem simulation, Baker et al. (2015) also found that the contributions from the BC to $PNO_3$ were overestimated when compared to IMPROVE data.



## 6. Summary and Discussion

The CAMx model and its PSAT source apportionment tool were used to examine and quantify the contributions of different source sectors and 27 source regions and the BC to the 2011 total inorganic $N_r$ deposition within the GYA. The source sectors were agriculture (AG), oil and gas activities (OG), wild and prescribed fires (Fire), and remaining contributions labeled as "Other". The Other sector was primarily composed of oxidized N originating from anthropogenic combustion

sources, including mobile and point sources, and the AG sector was almost entirely composed of reduced N compounds. Fires and the BC were a mix of reduced and oxidized N compounds. This assessment focused on only the inorganic N fraction. There is measured evidence that organic N (Prenni et al., 2014; Benedict et al., 2013a) is a significant contributor to $N_r$ deposition, and the inability to assess its origin in the current CTM is an important uncertainty in this work.

       Overall, the model simulation had a reasonable capacity to reproduce the measured seasonal and annual total $N_r$ deposition levels throughout the GYA. However, the model simulation underestimated available measured $NH_3$ concentrations by 65% on average, and measured $HNO_3$ concentrations were

overestimated by 108%. Therefore the model tended to overestimate the contribution of oxidized N compounds and underestimate those from reduced N compounds to the total $N_r$ deposition. In addition, both reduced and oxidized $N_r$ wet depositions were overestimated by 20–30%, which was due, at least partially, to the simulated precipitation frequency and magnitude being too high in the model. These biases suggest that the modeled contributions from the AG emission sector were underestimated while those from the Other sector's activities were overestimated.

       The simulated annual total $N_r$ deposition over the GYA in 2011 was 3.13 kg N ha$^{-1}$ yr$^{-1}$ and exceeded the CL estimates for lichen and lake chemistry

primarily at high elevation sites on the western slope and southern portion of the GYA. This finding is consistent with other studies using global models. Ellis et al. (2013) used the GEOS-Chem model to estimate the $N_r$ deposition to Class I areas for 2006 and showed that the simulated total $N_r$ deposition at GTNP (2.9 kg N ha$^{-1}$yr$^{-1}$) and YNP (2.6 kg N ha$^{-1}$yr$^{-1}$) exceeded the low end of CL for lichens (2.5 kg N ha$^{-1}$yr$^{-1}$).

       Emissions from the AG sector within the modeling domain were the largest contributor to the GYA total $N_r$ deposition budget at 34% per year. The contributions from the Other sector were also large at 26%. The OG emission sector generally had a small contribution, except at the southern edge of the GYA,

where it could contribute over 10% of the total $N_r$ deposition during winter months, with almost half of the OG contributions originating from emissions in the neighboring Jonah Field in western Wyoming. The Fire emission sector also had a significant contribution of 18% over the year. This was due to regional contributions from fires throughout the West and large contributions (>90%) at areas within the GYA where several wildfires occurred (Figure 7). The large impact from fires within the GYA is notable since the episodic nature of fire will result in differing year-to-year contributions from this uncontrollable sector.

       The largest impact from the AG emission sector originated from sources relatively close to the GYA, and the Snake River valley accounted for 74% of

the annual agricultural contribution. The agricultural contribution from Wyoming was 7%, and more-distant source regions in northern Utah, California, and the



northwestern United States each accounted for 4–5% of the agricultural contribution. Nearly half (45%) of the $N_r$ deposition from OG emission sector originated within Wyoming, especially the Upper Green River (27%). The largest impact from the Fire emission sector, including wildfire and prescribed fires, originated from Snake River valley (33%) and within the GYA (25%). The Other emission sector was more evenly distributed among near and distant regions, with the Snake River valley accounting for 23%, Wyoming 17%, and northern Utah, California, and the northwestern United States accounting for 14–16% of the $N_r$ deposition.

Long-range transport of N species from the BC, which primarily originated from international sources, contributed 21% of the total $N_r$ deposition within the GYA during 2011 and had the largest absolute contribution during the summer. Several studies have shown the importance of international source contributions to particulates and N deposition within the continental United States (Park et al., 2004; Brewer and Moore, 2009; Zhang et al., 2012; Fann et al., 2013; Baker et al., 2015; Thompson et al., 2015). However, the BC contribution in this work is on the high end of the reported values. For example, in a similar modeling study by Thompson et al. (2015), the estimated contribution of BC to $N_r$ deposition in Rocky Mountain National Park in 2009 was 13%. Zhang et al. (2012) used the GEOS-Chem model to evaluate N deposition in the United States during 2006–2008 and showed that foreign anthropogenic contributions were generally less than 10% but could rise up to 30% near the Canadian and Mexican borders. In addition, our evaluations of the BC suggest that the contribution of the BC to ambient $PNO_3$ and possibly other $N_r$ compounds was overestimated (Figure 11), clearly suggesting that more research is needed on the role of distant emission sources on impacting nitrogen deposition in remote areas, as well as further investigations into model biases.

The observed precipitation in 2011 was ~30–50% higher than the historical average (NOAA, 2012) with the largest bias occurring at the eastern sites in the GYA (Figure S9). This suggests that dry deposition of $NH_3$ may be a more important contributor to total $N_r$ deposition during spring than that observed during GrandTReNDS. Also, considering that the wet deposition in the GYA tended to be overestimated and the precipitation amount in 2011 was anomalously high, the source regions identified as having a higher weighting on the annual wet $N_r$ deposition budget (e.g., California) may not have such a significant impact as the current PSAT results suggested.

As discussed, source apportionment assessments of $N_r$ and its deposition to remote, ecologically sensitive areas such as the GYA have large uncertainties. Many of these uncertainties are known to the air quality modeling community, including the challenges of simulating precipitation in complex terrain, adequately characterizing $NH_3$ emissions from agricultural operations, the occurrence of wildfires, and the difficulty in simulating the $NH_3$ bi-directional flux and deposition flux of the other Nr compounds. Contributions from long-range transport of international emissions can also play a significant role in deposition in remote locations in the western United States. Further refinement in all of these areas is required to better understand and estimate the relative contributions of emission sources to excess N deposition within the GYA. Nevertheless, the modeling assessment showed that the reduced N contributed more



than 50% of the total $N_r$ deposition over the GYA, with >90% of the $NH_3$ emissions originating from agriculture sources. In addition, the Snake River valley in Idaho accounted for 74% of the agricultural contribution to the total $N_r$ deposition. Significant contributions from more-distant sources, e.g., California, and international sources to both the oxidized and reduced $N_r$ deposition illustrate the regional nature of the $N_r$ deposition problem. Emissions of oxidized N compounds are projected to continue to decrease, while emissions of ammonia are projected to remain relatively constant or increase (Li et al., 2016). This will

further increase the importance of the AG sector. However, exceedances of CL are still relatively small, and it is possible that decreased oxidized N deposition could reduce the $N_r$ deposition sufficiently to bring total Nr deposition below the CL in some GYA ecosystems.

**Acknowledgments**

This work was funded by the Air Resource Division of the National Park Service under cooperative agreement P17AC00773. The assumptions, findings, conclusions, judgments, and views presented herein are those of the authors and should not be interpreted as necessarily representing the National Park Service

policies. IMPROVE is a collaborative association of state, tribal, and federal agencies, and international partners. US Environmental Protection Agency is the primary funding source, with contracting and research support from the National Park Service. The Air Quality Group at the University of California, Davis is the central analytical laboratory, with ion analysis provided by Research Triangle Institute, and carbon analysis provided by Desert Research Institute. We acknowledge the Total Deposition (TDEP) Science Committee of the National Atmospheric Deposition Program (NADP) for their role in making the TDEP data and maps available. We also want to thank Helene Bennett from Cooperative Institute for Research in the Atmosphere (CIRA) at Colorado State University for

the technical editing of this research article.



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



**Figure captions**

**Figure 1.** Source region partition for CAMx PSAT simulation for the 12-km inner modeling domain (see Table S2 and Figure S2 for the details of the 27 source region partition) as well as locations of the monitoring sites at different networks ([a]Ammonia Monitoring Network; [b]Clean Air Status and Trends Network; [c]Grand Teton Reactive Nitrogen Deposition Study; [i]nteragency Monitoring of Protected Visual Environments; [e]Natiaonl Trend Network) used in the model performance evaluation of CAMx nitrogen species concentration and dry/wet deposition in the GYA (the black boundary line). The background map is the annual ammonia ($NH_3$) emission rate. The numbers in the figure are locations for the three sampling sites during GrandTReNDS and the 8 Class I areas in within the area: 1. Driggs, 2. Grand Targhee, 3. NOAA Climate Station station, 4. Grand Teton National Park, 5. John D. Rockfeller Jr. Memorial Parkway, 6. Yellowstone National Park, 7. Teton Wilderness, 8. Washakie Wilderness, 9. North Absaroka Wilderness, 10. Fitzpatrick Wilderness, and 11. Bridger Wilderness.

**Figure 2.** Model performance for (a-b) seasonal average $N_r$ concentration, (c-d) seasonal accumulated $N_r$ deposition budget as well as (e-f) seasonal accumulated precipitation amount at collocated location sites (YNP and Pinedale) over the GYA in 2011. [1]Clean Air Status and Trends Network; [2]Comprehensive Air Quality Model with extensions; [3]National Trend Network; [4]Parameter-elevation Relationships on Independent Slopes Model; [5]Weather Research and Forecasting model.

**Figure 3.** Inorganic nitrogen deposition budgets in absolute (Figure 3a) and in percentage (Figure 3c), as well as precipitation (Figure 3e), measured at the three core sites during the GrandTReNDS study period (April to September in 2011) with corresponding CAMx simulations (Figure 3b, Figure 3d, and Figure 3e). [1]Grand Teton Reactive Nitrogen Deposition Study; [2]Comprehensive Air Quality Model with extensions; [3]Weather Research and Forecasting model.

**Figure 4.** Annual nitrogen deposition budgets in absolute (Figure 4a) and in percentage (Figure 4b) as well as annual precipitation amounts (Figure 4c) from the NADP Total Deposition Map (TDEP) and corresponding CAMx (Comprehensive Air Quality Model with extensions) and WRF (Weather Research and Forecasting model) simulation results in 2011 at eight Class I areas across the GYA (the receptor sites on the x-axis are arranged from west to east in the GYA, see Figure 1. The reported CAMx dry and wet $N_r$ deposition values at the eight Class I areas are the average of the simulation values at corresponding grid cells for each area.).

**Figure 5.** Seasonal CAMx simulated $N_r$ deposition budgets averaged over the GYA in 2011. The left axis is the relative contribution of different $N_r$ species to seasonal $N_r$ deposition while the right axis is corresponding to the black diamonds for seasonal total $N_r$ deposition in absolute (kg N ha$^{-1}$).





**Figure 6.** Contributions of source sectors to the mean total $N_r$ deposition, dry $N_r$ deposition, and wet $N_r$ deposition over the GYA at different seasons in 2011. Figure 6a is the source sectors contributions in absolute and Figure 6b is the corresponding contributions in percentage.

**Figure 7.** Seasonal patterns of different source sectors' (agriculture, oil and gas activities, fires, others (e.g., anthropogenic, biogenic, lightning, and boundary conditions) contributions to total $N_r$ deposition over the GYA in 2011. The first column is the seasonal total $N_r$ deposition patterns in Kg N ha$^{-1}$ while the following five columns are the seasonal patterns of relative contributions from different source sectors.

**Figure 8.** Contributions of source regions to the mean total $N_r$ deposition, dry $N_r$ deposition, and wet $N_r$ deposition over the GYA at different seasons in 2011. Figure 8a is the source regions contributions in absolute and Figure 6b is the corresponding contributions in percentage.

**Figure 9.** Contributions of different source sectors as well as boundary conditions for total $N_r$ deposition in 2011 at 10 points of interest for critical load exceedance (see Table 2 for site locations and ecosystem impacts). The black-and-white pies are the contributions by source sectors while the color pies are the contributions by source regions. The color contour for the GYA boundary is the terrain heights with the legend at rightmost.

**Figure 10.** The sensitivity of $NH_3$ dry deposition velocity (left: "base" case, right: "DV_0.1" case with $NH_3$ dry deposition velocity slowing down) to source apportionment results over the GYA during July–August 2011. Figure 10a and 10c are the contributions by source regions in absolute and in percentage while Figure 10b and 10d are the contributions by source sectors.

**Figure 11.** The ratio of simulated versus measured particulate nitrate ($PNO_3$) concentrations against the boundary contributions to simulated $PNO_3$ at IMPROVE sites over a 12-km domain. [1]Model for Ozone and Related chemical Tracers; [2]Interagency Monitoring of Protect Visual Environments.



**Figure 1**

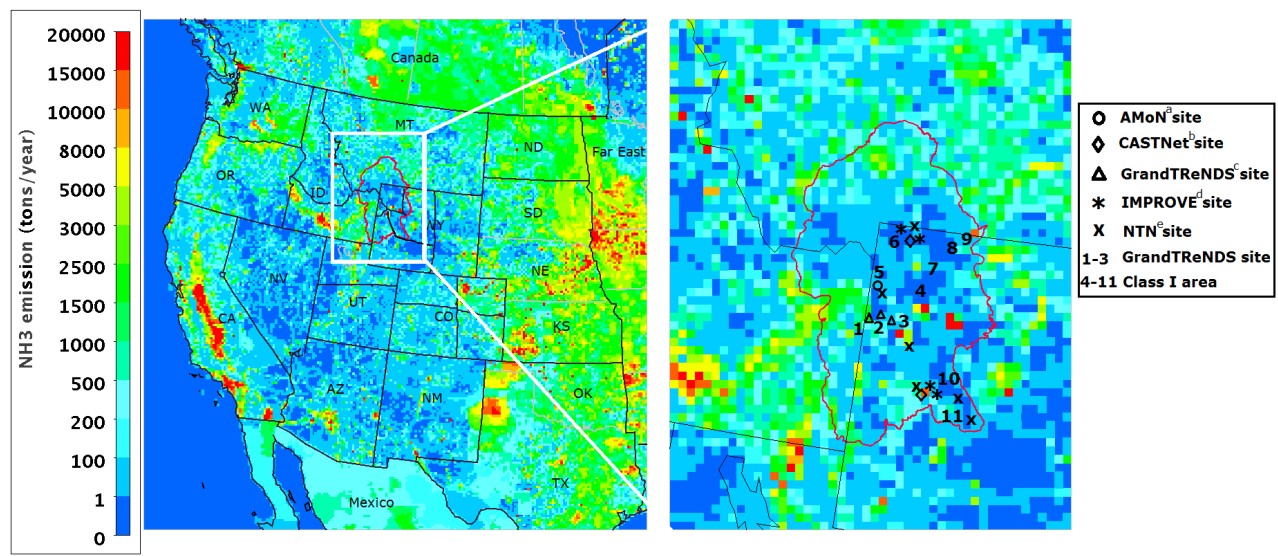





**Figure 2**

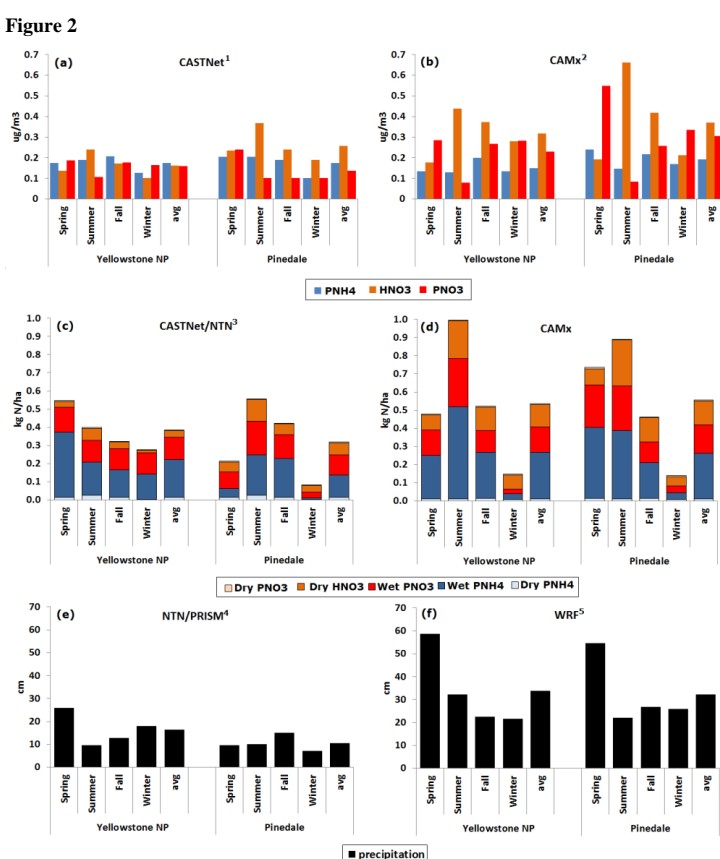



**Figure 3**

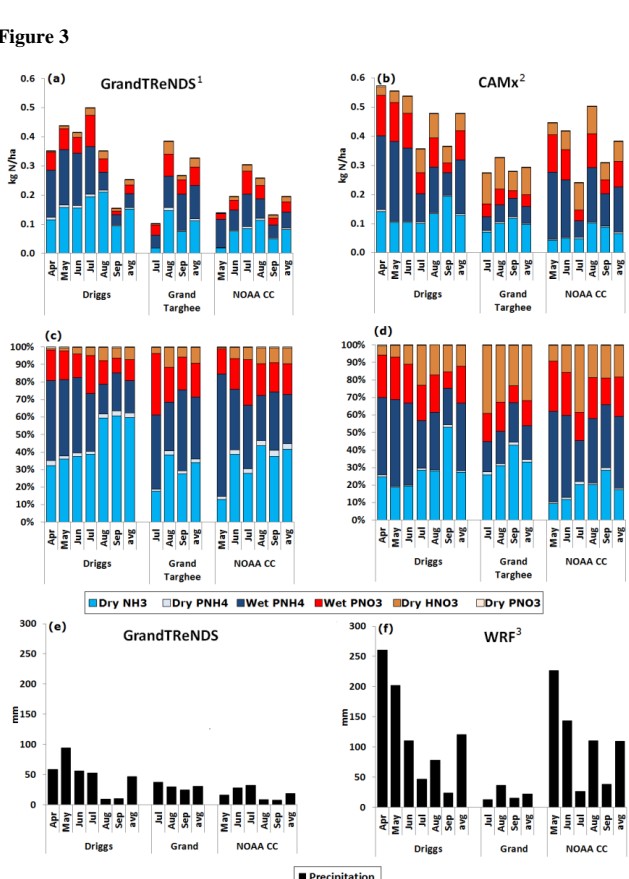





**Figure 4**

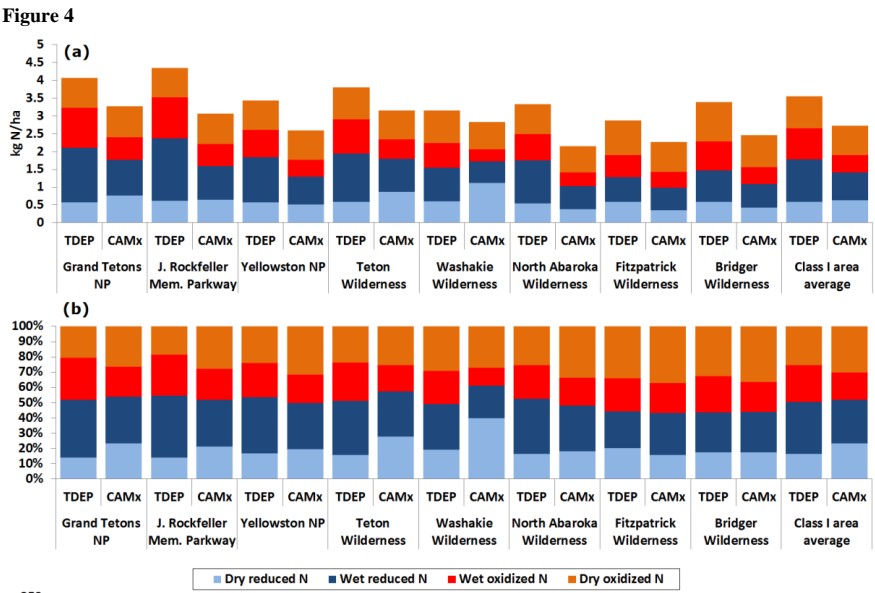

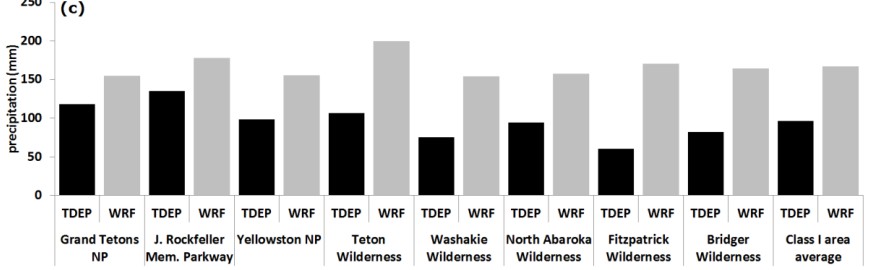





**Figure 5**

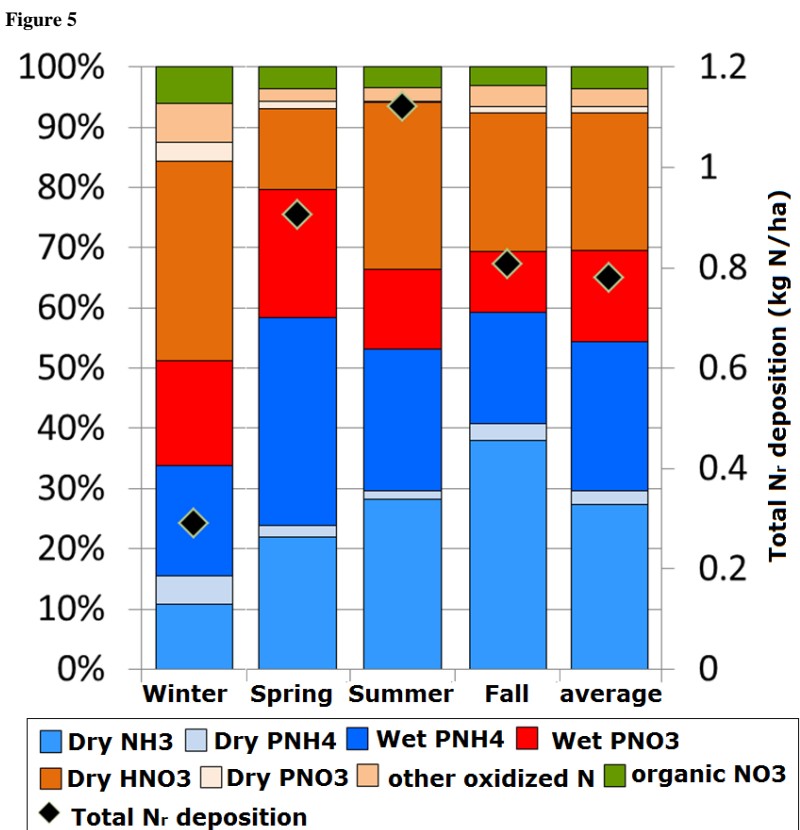



**Figure 6**

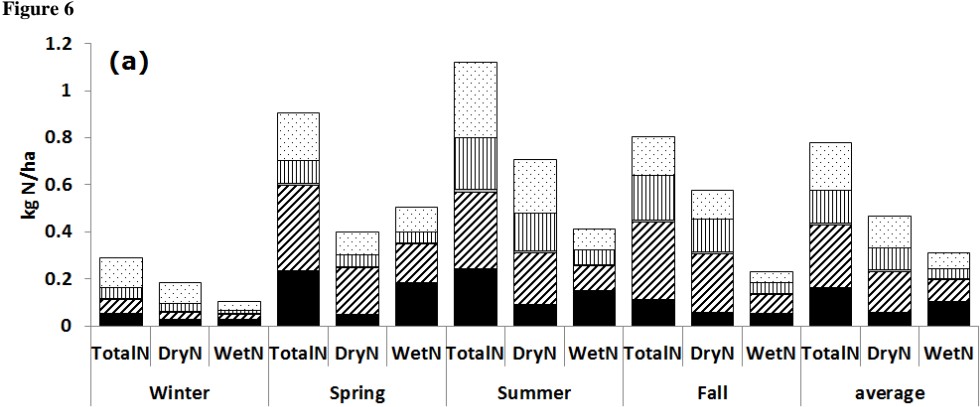

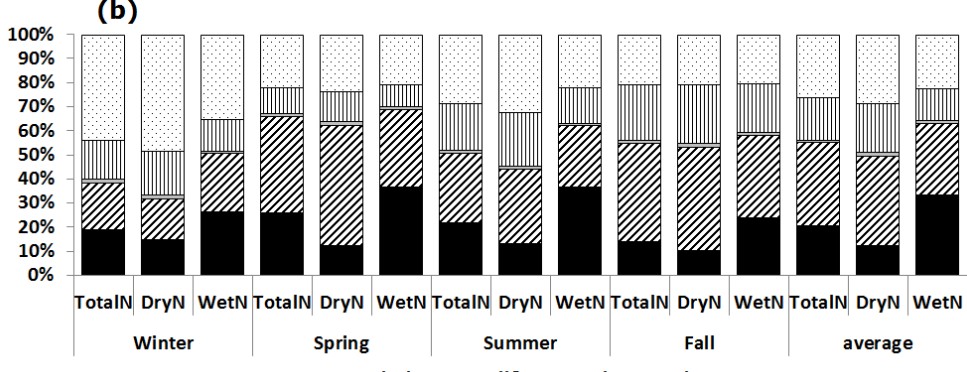



**Figure 7**

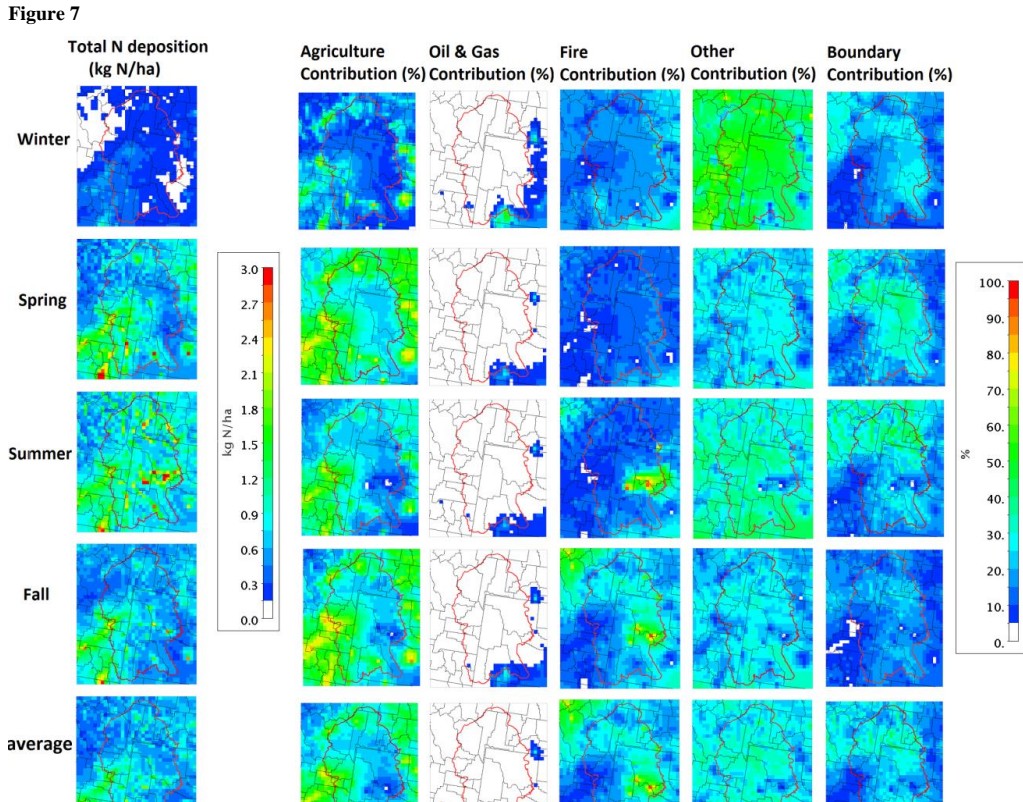





**Figure 8**

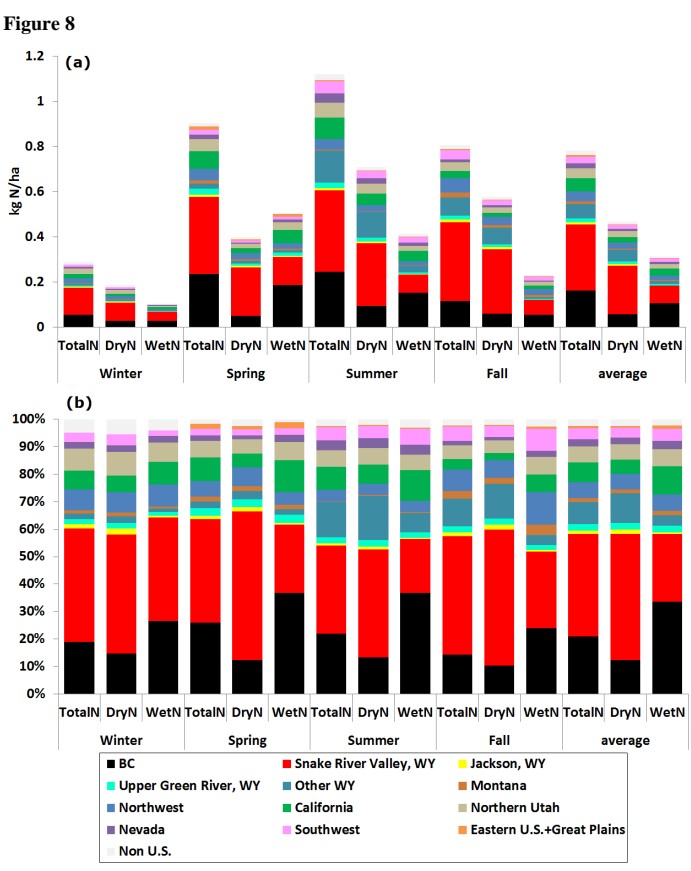





**Figure 9**

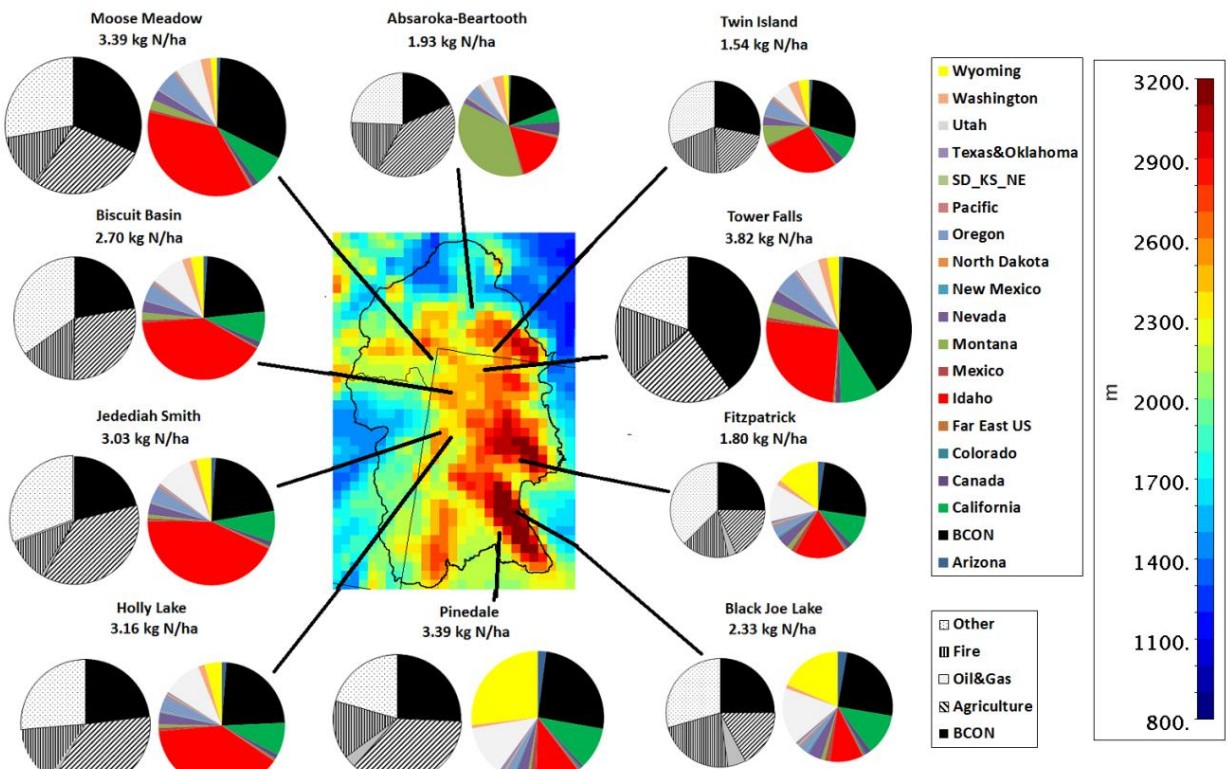




**Figure 10**

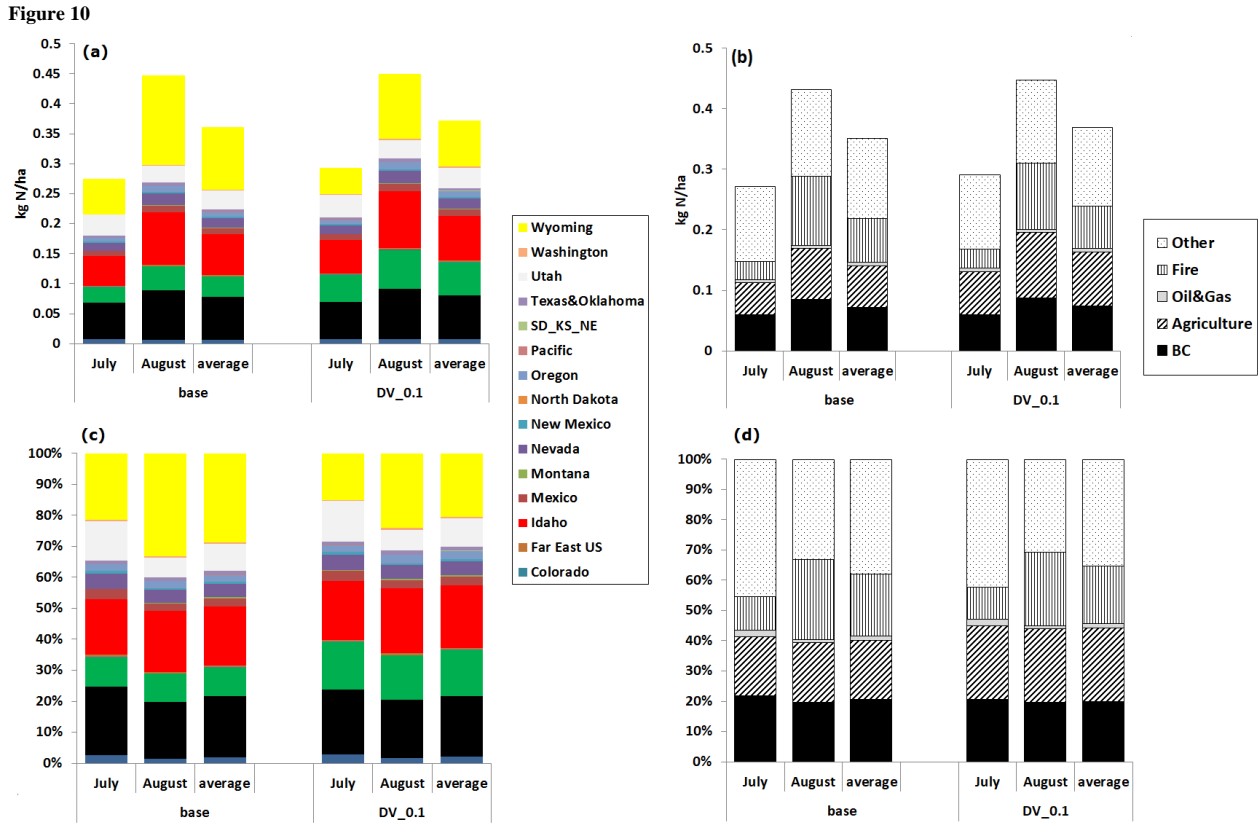



**Figure 11**

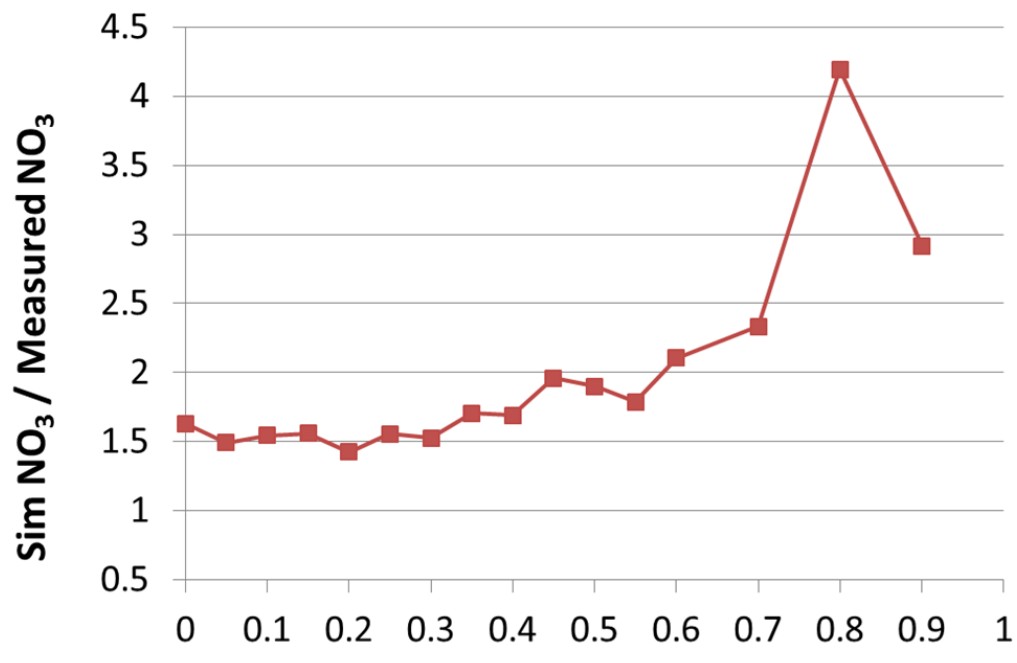





**Table 1.** Annual mean CAMx model performance for nitrogen species concentrations as well as nitrogen dry/wet depositions evaluated at sites in AMoN, CASTNet, IMPROVE, and NTN networks over the GYA region (see Figure 1 for site locations) in 2011.

| Species | | Network | Duration | OBS[a] | SIM[b] | #Site[c] | N[d] | R[e] | NMB[f] | NME[g] | FB[h] | FE[i] |
|---|---|---|---|---|---|---|---|---|---|---|---|---|
| concentration | $NH_3$ (ppb) | AMoN[1] | Sep 22-Dec 12, 2011 bi-weekly | 0.49 | 0.30 | 1 | 7 | 0.20 | -65% | 67% | -52% | 53% |
| | $HNO_3$ (ppb) | CASTNet[2] | Jan 4-Dec 27, 2011 weekly | 0.23 | 0.47 | 2 | 153 | 0.72 | 108% | 117% | 60% | 71% |
| | $PNO_3$ ($\mu g\ m^{-3}$) | CASTNet | Jan 4-Dec 27, 2011 weekly | 0.19 | 0.25 | 2 | 153 | 0.42 | 37% | 76% | 26% | 64% |
| | | IMPROVE[3] | Jan 3-Dec 29, 2011 every 3 days | 0.14 | 0.22 | 4 | 332 | 0.35 | 58% | 108% | 51% | 80% |
| | $PNH_4$ ($\mu g\ m^{-3}$) | CASTNet | Jan 4-Dec 27, 2011 weekly | 0.17 | 0.18 | 2 | 153 | 0.28 | 3% | 39% | 7% | 41% |
| N Deposition | $HNO_3$ dry (kg N ha$^{-1}$) | CASTNet | Jan 4-Dec 27, 2011 weekly | 0.0071 | 0.0187 | 2 | 153 | 0.81 | 153% | 156% | 77% | 82% |
| | PNO3 dry (kg N ha$^{-1}$) | CASTNet | Jan 4-Dec 27, 2011 weekly | 0.0012 | 0.0023 | 2 | 153 | 0.14 | 96% | 148% | 48% | 97% |
| | $PNH_4$ dry (kg N ha$^{-1}$) | CASTNet | Jan 4-Dec 27, 2011 weekly | 0.0018 | 0.0019 | 2 | 153 | 0.1 | 7% | 57% | 22% | 61% |
| | $NO_3^-$ wet (kg N ha$^{-1}$) | NTN[4] | Jan 4-Dec 27, 2011 weekly | 0.0079 | 0.0097 | 5 | 214 | 0.34 | 31% | 126% | 12% | 100% |
| | $NH_4^+$ wet (kg N ha$^{-1}$) | NTN | Jan 4-Dec 27, 2011 weekly | 0.0088 | 0.0126 | 5 | 214 | 0.32 | 49% | 142% | 19% | 106% |
| Precipitation (cm) | | NTN | Jan 4-Dec 27, 2011 weekly | 0.77 | 2.34 | 5 | 214 | 0.54 | 215% | 242% | 64% | 118% |

Note: [1]AMoN samples are collected for 2 weeks; [2]CASTNet samples are collected for 1 week; [3]IMPROVE 24-hr samples are collected every 3 days; [4]NTN samples are collected for 1 week; [a]average observation; [b]average simulation; [c]number of sites; [d]number of samples; [e]Pearson's correlation coefficient; [f]normalized mean bias; [g]normalized mean error; [h]fractional bias; [i]fractional errors.



**Table 2.** Total reactive nitrogen (Nr) deposition and critical loads for receptor points in the Greater Yellowstone Area in Wyoming.

| Site ID | Site Name (State) | Latitude /Longitude | Elevation (m) | Sensitive ecosystem | Total Nr deposition (kg N ha$^{-1}$) | | Critical load (kg N ha$^{-1}$)[3] | |
|---|---|---|---|---|---|---|---|---|
| | | | | | CAMx[1] | TDEP[2] | Range | confidence level |
| 1 | Absaroka-Beartooth Wilderness (MT) | 45.49 N 110.51 W | 2536 | Lichen | 1.93 | 2.80 | 3.02–4.89 | reliable |
| 2 | Twin Island (MT) | 45.07 N 109.81 W | 2829 | Lake chemistry | 1.53 | 3.99 | 2.5–7.1 | Fairly reliable |
| 3 | Tower Falls (WY) | 44.92 N 110.42 W | 2457 | Snowpack | 3.8 | 1.87 | 2.93–4.81[4] | reliable |
| 4 | Moose Meadow (ID) | 44.63 N 111.24 W | 1885 | Snowpack | 3.38 | 2.36 | 3.52–5.40[4] | reliable |
| 5 | Biscuit Basin (WY) | 44.46 N 110.83 W | 2050 | Snowpack | 2.69 | 3.49 | 3.39–5.27[4] | reliable |
| 6 | Jedediah Smith Wilderness (WY) | 43.79 N 110.94 W | 1944 | Lichen | 3.03 | 6.36 | 3.40–5.27 | reliable |
| 7 | Holly Lake (WY) | 43.79 N 110.79 W | 2230 | Lake chemistry | 3.15 | 5.50 | 2.5–7.1 | Fairly reliable |
| 8 | Fitzpatrick Wilderness (WY) | 43.40 N 109.66 W | 2890 | Lichen | 1.79 | 1.86 | 3.41–5.29 | reliable |
| 9 | Pinedale (WY) | 42.93 N 109.79 W | 2246 | Lichen | 3.39 | 2.67 | 2.66–4.53 | reliable |
| 10 | Black Joe Lake (WY) | 42.74 N 109.16 W | 3133 | Lake chemistry | 2.32 | 3.56 | 2.5–7.1 | Fairly reliable |

Note [1]Comprehensive Air Quality Model with extensions; [2]NADP Total Deposition maps. [3]The range of critical loads (CLs) for different effects on the selected sensitive ecosystem receptor is from United State CLAD (Critical Loads for Sulfur and Nitrogen Access Database), version 2.5 (Lynch et al., 2015). The level of confidence is based on the work of Pardo et al. (2011). The lower ends of the range were used in this study as a measured CL. [4]The CL values were for lichen response at sites with snow pack as a sensitive ecosystem.