# Peer review of "Source regions contributing to excess reactive nitrogen deposition in the Greater Yellowstone Area (GYA) of the United States"

_Atmospheric Chemistry and Physics, 2018_

## Referee Comment (RC1) · Anonymous Referee #1 · 25 May 2018

**General comments**

This manuscript reports on a modelling study, whereby the source sectors and regions of reactive nitrogen (Nr) are determined for the Greater Yellowstone Area in the United States. The model was evaluated thoroughly, and then used for quantifying source contributions to Nr deposition via a tagged model method. Agriculture from the Snake River Valley was determined to be the largest source. They took model error into account by doing a sensitivity study to give approximate uncertainties on the source contributions. This study represents new work as there is a lack of source attribution studies for Nr deposition for this region, however, I feel that they could emphasize

further how their study is new, different, and important compared to previous studies. Specific and technical comments below.

**Specific comments**

p2, line 21: state where the 40% of NH3 emissions from mobile applies? U.S. urban areas? A national average?

p4, first paragraph: can you emphasize more what's new from your study? It simply says that it "add to a growing body of Nr modeling source apportionment studies"? For example; is your study more detailed than that of Zhang et al (2012) and Lee et al (2016)? Does yours use a different technique (e.g., tagged model vs. zero-out scenario and adjoint model)? Is your study at higher resolution or does your model contain more detailed processes than GEOS-Chem? Etc. Emphasize why it was important to do this particular work despite the previous publications. Please also add to Section 6 to emphasize the importance of what's new in this study.

p4, line5: The sensitivity tests you did are an important part of this paper. I suggest emphasizing this more here in the introduction that this was done, given the large model biases.

p7, line 8: Comparing Table 1 in this paper to Figures 8, 11, and 12 in Simon et al, (2012), and it seems like CAMx model performance is within the range reported in Simon et al. However, just because it is within the range of what other models do, it doesn't necessarily follow that the model results are "adequate". Also the Simon et al. (2012) paper summarizes results published between 2006-2012, whereas model publications 2013-2017 may have improvements. Can you please add a few more recent references which have similar model biases as yours, and add some further justification to what is meant by "adequate"?

p10, line 12: it is mentioned above this that NH3 from agriculture is emitted into the first model layer and therefore doesn't get transported as far. Can you please also

discuss the fire emissions – specifically how high they get put into the model? It is described a bit on p4, lines 19-20, but can you mention here approximately how high the fires spread in the vertical, and thus how it would affect deposition at some distance downwind?

**Technical corrections**

p2, line 18: particulate nitrate (NO3), and other...

p6, line 22: may be related with the high...

p10, line 19: There is no "Table S4" in the supplement document. The table on the last page of the supplement has no label, and doesn't seem to be what you're talking about here. I think you may mean Table S3.

p24, line 4: (caption to Fig 1) National Trend Network: typo in National

p5, line 4: I expected to see the 24 tagged regions in Fig 1 given the text here, but actually that map is Fig S2. Text should be clarified. And I feel that knowing where those tagged regions are is important enough to be included in the main paper, rather than the supplemental material.

p.14, line 9-10: It wasn't *measured* HNO3 concentrations were overestimated by 108%. Modelled HNO3 was overestimated.

Fig 9: the Oil and Gas pattern is difficult to see in the legend – looks very similar to the Other pattern in the legend, and doesn't seem to be as dark as in the pies. In the pies, the Oil and Gas is (I think) the gray, but the legend looks much lighter. This doesn't seem to be a problem in Figs. 6 and 10 which has the same system.

Fig 11: I think the legend at the bottom should be removed because seeing MOZART/IMRPOVE next to the red square with the line through it is confusing and doesn't really make sense. It's not needed since in the text we know that the BC came from MOZART, and from the caption we know that the simulation was sampled at IM-

PROVE sites.

---

## Referee Comment (RC2) · Anonymous Referee #2 · 3 Jun 2018

The manuscript by Zhang et al. considers the sources of reactive nitrogen deposition in the Greater Yellowstone Area (GYA). The topic is timely and of relevance to this journal. The paper is in general clearly organized, well written, and is easy to read; the figures and tables are descriptive and appropriate. In terms of findings, the authors do a thorough job of first evaluating their modeling results compared to available measurements and other modeling studies in the literature. An issue is that they find very significant overestimation of HNO3 and underestimating of NH3. They then present source attribution results. Overall, findings of sources being from oxidized vs reduced nitrogen, different sectors, and different source reasons are interesting and seem sensible. They also consider a sensitivity study to try to address some of the modeling

shortcomings. My major criticism in this regard though is that such analysis or consideration of model biases is not reflected in the reporting of results elsewhere in the manuscript nor the abstract — given the rather significant model biases it seems results should be presented much more cautiously throughout. It would be useful if the authors could estimate some uncertainty ranges to their source attribution results — for example do they think they are accurate to within 1%? 10% an order of magnitude? Detailed comments along this line as well as a few other minor points are described in detail below. Addressing these would amount to minor revisions.

Specific comments:

Abstract: The model biases for NH3 and HNO3 are significant. Suggest adding some material to the abstract to address how modeled SA results should be interpreted, given these biases. Suggest referring to SA results as they pertain to the model (i.e., "largest source contributions in the model..."), unless this disconnect between measured and modeled values is resolved.

Abstract: importance of boundary conditions is not clear without having stated where these boundaries are. Nor is it clear that influence across the boundary would be international in origin (as opposed to natural oceanic emissions, recirculated domestic Nr, etc).

1.26: I thought is was already established that Nr deposition is already in excess (see first sentence of the abstract), thus it is odd here to say that the "results suggest that Nr deposition ...was above critical loads".

2.17: Worth indicating that these numbers are approximate and perhaps specific to a particular time period given trends in emissions from these sectors.

2.20: Missing some references here, e.g. work from Zondlo's group.

3.14: for zero-out –> using zero-out

3.17: "found the importance of emissions from California" is a bit vague. Were these

found to be more important than local sources? Or more important than otherwise expected?

3.19: This paragraph feels rather tangential and could be removed from the introduction or significantly shortened so only the content as it relates to understanding Nr dep in GYA.

4.13 - 20: several studies in the past year have identified an overestimation of mobile NOx emissions in the NEI2011 inventory. How were these addressed in the present work?

4.13 - 20: Does the inventory here contain the amount of NH3 from mobile sources mentioned in the introduction, or is if felt that this inventory under-represents this source?

4.13 - 20: It would be very useful for answering these questions and others if the emissions totals by sector and species for the different tagged regions could be included in the supporting information and summarized in the text (as opposed to the summaries mentioned in the introduction, which reflect values in the literature but do not specifically refer to the values used in the modeling for this work).

5.14: As anthropogenic SO2 emissions have declined in the US, the role of NOx and NH3 in forming ammonium nitrate aerosol has increased. How would PSAT account for the influence of the EGU sector via SO2 on deposition of PNH4 and PNO3, or is this not accounted for?

6.9: Could the authors clarify what constituted questionable data, such that their results could be more reproducible?

6.22: Does the mechanism for formation of N2O5 in CAMx match that in GEOS-Chem? If not, it's not clear how the reference to Heald et al. (2012) is relevant here.

7.2: Is a unidirectional NH3 emission model expected to lead to larger NH3 concentrations in this region of the US than a bidirectional flux model?

7.2: I would suspect that another possible factor leading to poor correlation and underestimation for NH3 is the overestimation of HNO3, which would promote excessive partitioning of NH3 to the particle phase. Did the authors consider evaluating NHx, or HNO3+PNO3, to get around the issues of partitioning (and thus hone in on issues related to sources and sinks)?

7.7: Are the performance metrics referenced here relevant for a study focusing on Nr source attribution? I could imagine if a studies goal was to forecast total PM2.5 concentrations, then opposing large biases in e.g. NH3 vs HNO3 would be of little concern; here, these issues seem much more considerable in terms of their impact on the final conclusions. Overall, I think the authors need to do more work in this regards to convince the readers of the merits of the application of the model so SA in the presence of such errors and biases.

Fig 3: I find it interesting that the measurements at each site show a distinct reduction in NH3 dry dep in September, whereas CAMx shows a maximum in September for Driggs and Grand Targhee. Can this authors comment on this?

12: How much did reducing the NH3 dry deposition change the total NH3 deposition amounts and their underestimation compared to observations mentioned in previous sections?

13.1: It seems like earlier their were several possible reasons for this, such as overestimated HNO3 concentrations, and yet here only precipitation biases are considered?
* * *

---

## Author Comment (AC1) · 4 Aug 2018

*General comments*

*This manuscript reports on a modelling study, whereby the source sectors and regions of reactive nitrogen (Nr) are determined for the Greater Yellowstone Area in the United States. The model was evaluated thoroughly, and then used for quantifying source contributions to Nr deposition via a tagged model method. Agriculture from the Snake River Valley was determined to be the largest source. They took model error into account by doing a sensitivity study to give approximate uncertainties on the source contributions. This study represents new work as there is a lack of source attribution studies for Nr deposition for this region, however, I feel that they could emphasize further how their study is new, different, and important compared to previous studies.*

**Response:**

Thanks for the recognition of the value of this modeling study and providing the opportunity for us to revise the manuscript accordingly. In order to emphasize the importance and new findings compared with previous modeling studies targeting nitrogen deposition in remote areas of the United States, we follow the suggestions of the reviewer to add a few sentences to emphasize how our study stands out compared with previous similar source apportionments. The detailed changes can be seen in the "track changes" version of the revised manuscript as well as in the responses to the specific comments below.

*Specific and technical comments below.*

*Specific comments*

*p2, line 21: state where the 40% of NH3 emissions from mobile applies? U.S. urban areas? A national average?*

**Response:**

The sentence: "Mobile sources are also an important source of NH3 and can be the primary emitter in urban areas. A recent study found the increasing importance of on-road emissions of NH3, which at 40% exceed agricultural emissions (Fenn et al., 2018)."
Was modified to:

"Mobile sources are also an important source of NH3 and can be the primary emitter in urban areas (Sun et al., 2014; Sun et al., 2017). Emissions from this sector have large uncertainties and a recent study suggests that on-road NH3 emissions in the 2011 National Emissions Inventory (NEI) were underestimated by a factor of 2.9 (Fenn et al., 2018)."

*p4, first paragraph: can you emphasize more what's new from your study? It simply says that it "add to a growing body of Nr modeling source apportionment studies"? For example; is your study more detailed than that of Zhang et al (2012) and Lee et al (2016)?*

*Does yours use a different technique (e.g., tagged model vs. zero-out scenario and adjoint model)? Is your study at higher resolution or does your model contain more detailed processes than GEOS-Chem? Etc. Emphasize why it was important to do this particular work despite the previous publications. Please also add to Section 6 to emphasize the importance of what's new in this study.*

**Response:**

Based on the reviewer's suggestion, we revised this paragraph to the following:

"In this work, we add to the growing body of Nr modeling source apportionment studies by conducting a detailed analysis using the Particulate Source Apportionment Technology (PSAT) module within the CAMx (Comprehensive Air Quality Model with extensions) (Ramboll Environ, 2014) CTM to quantify the seasonal contributions from different source regions and source sectors to Nr throughout the GYA. Compared with previous Nr deposition simulation studies in United States, this work uses tagged reactive tracers to attribute the contributions from four designated emission sectors and 27 designated emission regions to Nr deposition in the GYA with a much higher horizontal grid resolution (12 km) and an up-to-date emission inventory instead of using a zero-out approach (e.g., Zhang et al., 2012) or an adjoint model (e.g., Lee et al., 2016). The model simulation of Nr and its constituents were first evaluated against routine measured data as well as the unique data measured during the GrandTReNDS campaign period (Benedict et al., 2013a; Prenni et al., 2014). Nr deposition from CAMx simulations was also compared with total deposition maps (TDEP), which were developed for deposition trend analysis and ecological impact assessment (Schwede and Lear, 2014). The detailed source apportionment results are presented here, focusing on seasonal variations and the relative importance to CL exceedance in sensitive ecosystems within the GYA. The discussion of identified model bias and uncertainties to source apportionment results interpretation, including the model lateral boundary conditions, the impact of model precipitation to wet deposition simulation, and the impact of ammonium dry deposition velocity to dry deposition are also presented."

Also, in section 6, the first paragraph, we added a sentence to emphasize the uniqueness or the importance of our modeling work here:

"Nevertheless, this $N_r$ source apportionment work is the first thorough analysis of the origin of inorganic Nr in the GYA using a regional air quality modeling platform. The detailed source sector and source region configurations in PSAT enabled quantitative, though uncertain, estimates of their relative importance. This is needed information by stakeholder and regulator groups to understand the causes of excess Nr deposition in the GYA, monitor changes in Nr deposition and develop possible future mitigation strategies"

*p4, line5: The sensitivity tests you did are an important part of this paper. I suggest emphasizing this more here in the introduction that this was done, given the large model biases.*

**Response:**

We changed the sentence from "The final source apportionment results are then interpreted within the context of the identified model bias and uncertainties" to "The discussion of identified model bias and uncertainties to source apportionment results interpretation, including the model lateral boundary conditions, the impact of model precipitation to wet deposition simulation, and the impact of ammonium dry deposition velocity to dry deposition are also presented"

*p7, line 8: Comparing Table 1 in this paper to Figures 8, 11, and 12 in Simon et al, (2012), and it seems like CAMx model performance is within the range reported in Simon et al. However, just because it is within the range of what other models do, it doesn't necessarily follow that the model results are "adequate". Also the Simon et al. (2012) paper summarizes results published between 2006-2012, whereas model publications 2013-2017 may have improvements. Can you please add a few more recent references which have similar model biases as yours, and add some further justification to what is meant by "adequate"?*

**Response:**

We do not explicitly use the word "adequate" in the description of the base model performance from CAMx in 2011. As requested, we added additional citations from the model publications from 2013 to 2017 with similar model biases to justify that the modeling platform we were working with has the capability to capture the general spatial and temporal variations of the reactive nitrogen in the atmosphere and that the model performance is in line with the peer modeling results applied for the continental United States using regional photochemical models (e.g., CMAQ and CAMx). Also, we provided Table S3 in the supplementary material to summarize model performance of series simulations with nitrogen-deposition-related species.

We deleted the sentence referring only to the Simon et al. (2012) study and added the new description at the end of this section as follows:

"Table S3 provides a comparison of regional air quality model, N- related species performance, evaluated by observations over the United States from peer-reviewed studies in recent years (e.g., Simon et al., 2012; Bash et al., 2013; Zhang et al., 2013; Yu et al., 2014; Thompson et al., 2015; Li et al., 2017), and it shows that our results are comparable, with some similar model biases such as overestimation of HNO3 and underestimation of NH3. Overall, the CAMx results provide a reasonable platform for evaluation of the contribution of sources to Nr deposition throughout the GYA."

Table S3. Summary of regional air quality model nitrogen related species performance in terms of normalized mean bias (NMB) evaluated by observations over the continental United States

| Species | Photochemical model | Duration | Model resolution | Region evaluated | NMB value | Reference |
|---|---|---|---|---|---|---|
| NH3 | CAMx | 2011 full year | 12km | GYA | -65% | this study |
| | CMAQ,CAMx | Jan, Jul 2002 | 4km | Southwest US | [-23% -79%] | Zhang et al. (2013) |
| | CAMx | 2009 full year | 36km/12km | Colorado | -55% | Thompson et al. (2015) |
| | CAMx | Summer 2011 | 4km | Colorado | [-83% 46%] | Li et al. (2017) |
| HNO3 | CAMx | 2011 full year | 12km | GYA | 108% | this study |
| | CAMx | 2009 full year | 36km/12km | Colorado | 23% | Thompson et al. (2015) |
| | CMAQ,CAMx | Jan, Jul 2002 | 4km | Southwest US | [-17% 45%] | Zhang et al. (2013) |
| PM25 nitrate | CAMx | 2011 full year | 12km | GYA | [37% 58%] | this study |
| | CAMx,CMAQ, WRF-Chem (n=34) | varies, case study to full year simulation | 9-45km | varies, states to CONUS | [-49% 11%] | Simon et al. (2012) |
| | CMAQ,CAMx | Jan, Jul 2002 | 4km | Southwest US | [-92% -103%] | Zhang et al. (2013) |
| | CAMx | 2009 full year | 36km/12km | Colorado | 57% | Thompson et al. (2015) |
| | CMAQ | 2002 full year | 12km | CONUS | [-24% 45%] | Bash et al. (2013) |
| | CMAQ | 1990-2010 | 108km | CONUS | [-41% 106%] | Xing et al. (2015) |
| | WRF-CMAQ | Aug, Sep 2006 | 12/4km | CONUS/Texas | [-82% 83%] | Yu et al. (2014) |
| PM25 ammonia | CAMx | 2011 full year | 12km | GYA | 3% | this study |
| | CMAQ,CAMx | Jan, Jul 2002 | 4km | Southwest US | [-57% 61%] | Zhang et al. (2013) |
| | CAMx,CMAQ, WRF-Chem (n=31) | varies, case study to full year simulation | 9-45km | varies, states to CONUS | [-17% 7%] | Simon et al. (2012) |
| | CAMx | 2009 full year | 36km/12km | Colorado | -31% | Thompson et al. (2015) |
| | CMAQ | 1990-2010 | 108km | CONUS | [-54% 23%] | Xing et al. (2015) |
| | WRF-CMAQ | Aug, Sep 2006 | 12/4km | CONUS/Texas | [-75% 48] | Yu et al. (2014) |
| NOx wet deposition | CAMx | 2011 full year | 12km | GYA | 31% | this study |
| | CAMx,CMAQ, WRF-Chem (n=16) | varies, case study to full year simulation | 9-45km | varies, states to CONUS | [-45% 19%] | Simon et al. (2012) |
| | CMAQ,CAMx | Jan, Jul 2002 | 4km | Southwest US | [-94% 52%] | Zhang et al. (2013) |
| NH4 wet deposition | CAMx | 2011 full year | 12km | GYA | 49% | this study |
| | CAMx,CMAQ, WRF-Chem (n=16) | varies, case study to full year simulation | 36km/12km | varies, states to CONUS | [-33% 28%] | Simon et al. (2012) |
| | CMAQ,CAMx | Jan, Jul 2002 | 4km | Southwest US | [-51% 19%] | Zhang et al. (2013) |
| | CMAQ | 2002 full year | 12km | CONUS | [-16% 18%] | Bash et al. (2013) |

We also add the following citations to the reference list:

Bash, J.O., Cooter, E.J., Dennis, R.L., Walker, J.T., and Pleim, J.E. (2013), Evaluation of a regional air-quality model with bidirectional NH3 exchange coupled to an agroecosystem model, Biogeoscience, 10, 1635-1645, doi:10.5194/bg-10-1635-2013.

Li, Y., Thompson, T.M., Damme, M.V., Chen, X., Benedict, K.B., Shao, Y., Day, D., Boris, A., Sullivan, A.P., Ham, J. and Whitburn, S.: Temporal and spatial variability of ammonia in urban and agricultural regions of northern Colorado, United States, Atmos. Chem. Phys., 17(10), 6197-6213, 2017.

Thompson, T.M., Rodriguez, M.A., Barna, M.G., Gebhart, K.A., Hand, J.L., Day, D.E., Malm, W.C., Benedict, K.B., Collett, J.L. and Schichtel, B.A.: Rocky Mountain National Park reduced nitrogen source apportionment, J. Geophys. Res., 120(9), 4370-4384, 2015.

Xing, J., Mathur, R., Pleim, J., Hogrefe, C., Gan, C.M., Wong, D.C., Wei, C., Gilliam, R. and Pouliot, G., Observations and modeling of air quality trends over 1990–2010 across the Northern Hemisphere: China, the United States and Europe. Atmos. Chem. Phys., 15, 2723-2747, 2015.

Yu, S., Mathur, R., Pleim, J., Wong, D., Gilliam, R., Alapaty, K., Zhao, C. and Liu, X., Aerosol indirect effect on the grid-scale clouds in the two-way coupled WRF-CMAQ: model description, development, evaluation and regional analysis, 14, 11247-11285, Atmos. Chem. Phys., 2014.

Zhang, Y., Olsen, K.M. and Wang, K., Fine scale modeling of agricultural air quality over the southeastern United States using two air quality models. Part I. Application and evaluation. Aerosol Air Qual. Res., 13(4), 1231-1252, 2013.

*p10, line 12: it is mentioned above this that NH3 from agriculture is emitted into the first model layer and therefore doesn't get transported as far. Can you please also discuss the fire emissions – specifically how high they get put into the model? It is described a bit on p4, lines 19-20, but can you mention here approximately how high the fires spread in the vertical, and thus how it would affect deposition at some distance downwind?*

**Response:**

We used the fire emissions developed from the Particulate Matter Deterministic and Empirical Tagging and Assessment of Impacts on Levels (PMDETAIL) study (Moore et al., 2012). The emissions for fire activities include prescribed fires and wildfires. In the PMDETAIL fire plume rise methodology (Mavko and Morris, 2013), three parameters were defined to provide the release heights of fire smoke emissions as hourly inputs to CAMx, namely (1) height above ground of plume top ($P_{top}$), (2) height above ground of plume bottom ($P_{bot}$), and (3) the fraction of emissions emitted near the ground ($f_{Lay1}$). When allocating the fire emissions to different vertical layers according to the CAMx vertical layer setting, the PMDETAIL methodology included the WRF estimated hourly planetary boundary layer (PBL) in the grid cell containing the fire emissions and injected the fire emissions near the surface between the CAMx model layer 1 and the maximum of $P_{bot}$ and PBL values:

Fire emission ($f_{Lay1}$) = ground to max.($P_{bot}$, PBL)

For the elevated fire emissions, the PMDETAIL methodology released the emissions in layers between $P_{bot}$ and the maximum of $P_{top}$ and PBL value for the hour and grid cell of the fire:

Fire emission ($1-f_{Lay1}$) = $P_{bot}$ to max.($P_{top}$, PBL)

We did not have the detailed information for those three parameters for each fire accounted for in the PMDETAIL and used in the 2011 CAMx modeling. However, looking at the attached figure below, we can deduce that those three fire plumes in summer within the GYA were injected into the vertical layer between $P_{bot}$ and the model PBL height so that it may be mostly mixed within the PBL and has the dominant impact to adjunct grids where the fire emission occurs. It has little chance to disperse higher and impact N deposition at a longer distance downwind.

We changed page 4, lines 19–20, from "PMDETAIL developed 2011 fire emissions using satellite data, ground detects, and burn scar and estimated the plume rise depending on fire size and type (Mavko and Morris, 2013)." to "PMDETAIL developed 2011 fire emissions using satellite data, ground detects, and burn scar and estimated the plume rise, depending on fire size and type. The hourly, nonsurface fire emissions were allocated to the proper CAMx vertical layers based on the model-predicted planetary boundary layer (PBL) height and the spanning of the plume top and bottom above the ground (Mavko and Morris, 2013)."

We added Figure S4 to the supplementary file to show that the fires occurring during summer 2011 near the GYA predominantly impacted the adjacent grids. Now the sentences on page 12, line 12 that describe the fire emission impact to seasonal N deposition in the GYA read as "The footprint of fire emission impacts depends on the simulated injection height of the fire plumes. The emissions from fires that occurred within the GYA during the summer and fall likely remained within the mixed layer and had less chance to be transported far downwind to impact more distance areas (Figure S4)."

[Figure]

Figure S4. (left) Spatial pattern of total NOx emission from Fire emission sectors during summer (June, July, August) 2011 near the Greater Yellowstone Area (GYA). (right) the Spatial pattern of total N deposition attributed to Fire emission during summer 2011.

References:

   Mavko, M. and Morris, R., DEASCO3 project updates to the fire plume rise methodology to model smoke dispersions. Air Science Inc. Portland, Oregon and ENVIRON International Corporation, Novato, California. December 3, 2013. http://wraptools.org/pdf/DEASCO3_Plume_Rise_Memo_20131210.pdf

Moore, C.T., Randall, D., Mavko, M., Morris, R., Koo, B., Fitch, M., George, M., Barna, M., Vimont, J., Anderson, B. and Acheson A., Deterministic and empirical assessment of smoke's contribution to ozone (DEASCO3), final report, 2012, Joint Fire Science, Program Project #11-1-6-6, https://www.firescience.gov/projects/11-1-6-6/project/11-1-6-6_final_report.pdf.

*Technical corrections*

*p2, line 18: particulate nitrate (NO3), and other...*

**Response:**

To be consistent with the notation in other places in the manuscript, such as page 5, line 14, and Table 1, we changed the sentence from "Atmospheric reactions of NOx result in nitric acid (HNO3), particulate nitrate, and other compounds." to "Atmospheric reactions of NOx result in nitric acid (HNO3), particulate nitrate (PNO3), and other compounds."

*p6, line 22: may be related with the high: : :*

**Response:**

Changed from "The poor NH3 results may related with the high …" to "The poor NH3 results may be related to the high …".

*p10, line 19: There is no "Table S4" in the supplement document. The table on the last page of the supplement has no label, and doesn't seem to be what you're talking about here. I think you may mean Table S3.*

**Response:**

We corrected the sentence to "Most (74%) of the Nr from this region was from the AG source sector and was composed of reduced N (Table S3)." The last table in the supplemental material belongs with the supplementary File S1 in the section "regional evaluation of CAMx nitrogen deposition in 2011" and is therefore not assigned a label.

*p24, line 4: (caption to Fig 1) National Trend Network: typo in National*

**Response:**

Corrected the typo from "Natiaonl" to "National".

*p5, line 4: I expected to see the 24 tagged regions in Fig 1 given the text here, but actually that map is Fig S2. Text should be clarified. And I feel that knowing where those tagged regions are is important enough to be included in the main paper, rather than the supplemental material.*

**Response:**

We followed the suggestion to move the Figure S2, including the 27 tagged regions, from supplemental material into the main content. The caption in old Figure 1 (now Figure 2) has been changed to clarify that the source region partition for the CAMx PSAT

simulation shown here is only for the 12-km inner modeling domain. The number of the figures in main document and supplemental material has changed accordingly.

*p.14, line 9-10: It wasn't measured HNO3 concentrations were overestimated by 108%. Modelled HNO3 was overestimated.*

**Response:**

Changed the sentence from "However, the model simulation underestimated available measured NH3 concentrations by 65% on average, and measured HNO3 concentrations were overestimated by 108%." to "However, the model simulation underestimated the measured NH3 concentrations by 65% on average and overestimated the measured HNO3 by 108%."

*Fig 9: the Oil and Gas pattern is difficult to see in the legend – looks very similar to the Other pattern in the legend, and doesn't seem to be as dark as in the pies. In the pies, the Oil and Gas is (I think) the gray, but the legend looks much lighter. This doesn't seem to be a problem in Figs. 6 and 10 which has the same system.*

**Response:**

We double-checked Figure 10 (previously Figure 9) and made sure the legend, color map setting, as well as notation are consistent with Figure 7 (previously Figure 6) and Figure 11 (previously Figure 10). The updated Figure 9 is attached here for reference.

[Figure]

*Fig 11: I think the legend at the bottom should be removed because seeing MOZART/IMRPOVE next to the red square with the line through it is confusing and doesn't really make sense. It's not needed since in the text we know that the BC came from MOZART, and from the caption we know that the simulation was sampled at IMPROVE sites.*

**Response:**

We accepted this suggestion to revise the caption for Figure 12 (previously Figure 11) as: "Figure 12. Ratio of simulated versus measured particulate nitrate (PNO3) concentrations against the boundary contributions to simulated PNO3 at IMPROVE sites over a 12-km domain."

The revised figure attached below.

[Figure]

---

## Author Comment (AC2) · 4 Aug 2018

_Comments from anonymous referee #2_

_General comments_

_The manuscript by Zhang et al. considers the sources of reactive nitrogen deposition in the Greater Yellowstone Area (GYA). The topic is timely and of relevance to this journal. The paper is in general clearly organized, well written, and is easy to read; the figures and tables are descriptive and appropriate. In terms of findings, the authors do a thorough job of first evaluating their modeling results compared to available measurements and other modeling studies in the literature. An issue is that they find very significant overestimation of HNO3 and underestimating of NH3. They then present source attribution results. Overall, findings of sources being from oxidized vs reduced nitrogen, different sectors, and different source reasons are interesting and seem sensible. They also consider a sensitivity study to try to address some of the modeling shortcomings._

_My major criticism in this regard though is that such analysis or consideration of model biases is not reflected in the reporting of results elsewhere in the manuscript nor the abstract given the rather significant model biases it seems results should be presented much more cautiously throughout. It would be useful if the authors could estimate some uncertainty ranges to their source attribution results at for example do they think they are accurate to within 1%? 10% an order of magnitude? Detailed comments along this line as well as a few other minor points are described in detail below. Addressing these would amount to minor revisions._

**Response:**

We appreciate the favorable overall sentiment and the opportunity to revise our manuscript in response to those comments. We have addressed each comment and suggestion as described below. Note that we do not know the uncertainties in the source attribution (SA) results, but suspect that they are large based on the model performance evaluation. This is why the results are discussed in more general and semi-quantitative terms in section 5. However, in response to the comment we have made a greater effort to convey the uncertainties and potential biases where appropriate. For example, in the abstract we included the sentences: "These uncertainties appear to result in an overestimation of distant source regions including California and BC and an underestimation of closer agricultural source regions including the Snake River valley. Due to these large uncertainties the relative contributions from the modelled sources and their general patterns are the most reliable results."

Also, the discussions on the change of deposition velocity of NH3 in CAMx to SA results showed that less than 10% change of the contributions for each source sectors/regions for the conducted 2 month sensitivity simulations (Figure 11). Also, the SA results due to different boundary conditions usage didn't change much (less than 10%, see Figure S8). The detailed comment below further address this issue.

*Specific comments:*

*Abstract: The model biases for NH3 and HNO3 are significant. Suggest adding some material to the abstract to address how modeled SA results should be interpreted, given these biases. Suggest referring to SA results as they pertain to the model (i.e., "largest source contributions in the model. . .), unless this disconnect between measured and modeled values is resolved.*

**Response:**

We agree with the reviewer and added the following sentences to the abstract:

"These uncertainties appear to result in an overestimation of distant source regions including California and BC and an underestimation of closer agricultural source regions including the Snake River valley. Due to these large uncertainties the relative contributions from the modelled sources and their general patterns are the most reliable results."

*Abstract: importance of boundary conditions is not clear without having stated where these boundaries are. Nor is it clear that influence across the boundary would be international in origin (as opposed to natural oceanic emissions, recirculated domestic Nr, etc).*

**Response:**

The following sentence was added to the abstract: "The BC were outside the conterminous United States and thought to represent international anthropogenic and natural contributions."

*1.26: I thought it was already established that Nr deposition is already in excess (see first sentence of the abstract), thus it is odd here to say that the "results suggest that Nr deposition ...was above critical loads".*

**Response:**

We deleted this sentence as suggested.

*2.17: Worth indicating that these numbers are approximate and perhaps specific to a particular time period given trends in emissions from these sectors.*

**Response:**

Based on the suggestion, this sentence now read as:

"These compounds arise from a variety of sources, with inorganic oxidized N primarily emitted as nitrogen oxides (NOx) from fossil fuel combustion, with approximately 25% from power plants, 50% from automobiles, and 10% from other mobile sources on annual based county level estimation (EPA, 2015)."

*2.20: Missing some references here, e.g. work from Zondlo's group.*
**Response:**
We added two highly cited references from Zondlo's group regarding the on-road NH3 emissions (Sun et al., 2014; Sun et al., 2017). The sentence now read as
"Mobile sources are also an important source of NH3 and can be the primary emitter in urban areas (Sun et al., 2014; Sun et al., 2017)."

References:
Sun, K., Tao, L., Miller, D.J., Khan, M.A. and Zondlo, M.A.: On-road ammonia emissions characterized by mobile, open-path measurements. Environ. Sci. Tech., 48(7), 3943-3950, 2014.
Sun, K., Tao, L., Miller, D.J., Pan, D., Golston, L.M., Zondlo, M.A., Griffin, R.J., Wallace, H.W., Leong, Y.J., Yang, M.M. and Zhang, Y. Vehicle emissions as an important urban ammonia source in the United States and China. Environ. Sci. Tech., 51(4), 2472-2481, 2017.

*3.14: for zero-out –> using zero-out*
**Response:**
Changed.

*3.17: "found the importance of emissions from California" is a bit vague. Were these found to be more important than local sources? Or more important than otherwise expected?*
**Response:**
Lee et al. (2016) used the adjoint of GOES-Chem to investigate the spatial and sectoral distribution of annual Nr deposition contributed by different sources. As expected, NH3 emissions from livestock and NOx emissions from mobile sources are the major contributors to Nr deposition in nearly all selected Class I areas in the United States. Nr deposition in the mountain regions in the western U.S (Grand Teton and Rocky Mountain NPs) are ~50% from nearby sources (<400 km) and the rest from sources as far away as California (~1300 km). To avoid the ambiguity, we rewrote this sentence as:

"Lee et al. (2016) used the adjoint version of GEOS-Chem to quantify the sources of Nr deposition in eight selected federal Class I areas in 2010 and found a nonnegligible footprint (>20%) of Nr deposition in western United States, including GTNP and Rocky Mountain National Park (RMNP), attributed to long-range transport from sources in California, especially during summer time."

Reference:

Lee, H. M., Paulot, F., Henze, D. K., Travis, K., Jacob, D. J., Pardo, L. H., and Schichtel, B. A.: Sources of nitrogen deposition in Federal Class I areas in the US, Atmos. Chem. Phys., 16(2), 2016.

*3.19: This paragraph feels rather tangential and could be removed from the introduction or significantly shortened so only the content as it relates to understanding Nr dep in GYA.*
**Response:**
We significantly shortened this paragraph into one sentence and combined it with the previous paragraph to show the similarity of source apportionment modeling studies' focus on Rocky Mountain to the GYA area. Now the new sentence read as:
"Similar modeling studies focusing on RMNP also suggested the important contributions of distant sources including those from California and other counties and the fact that the contributions from source of reduced Nr were larger than those from sources of oxidized Nr (Thompson et al., 2015; Malm et al., 2016)."

*4.13 - 20: several studies in the past year have identified an overestimation of mobile NOx emissions in the NEI2011 inventory. How were these addressed in the present work?*
**Response:**
The mobile emissions we used in this modeling study were from the NEI 2011 inventory, which used MOVES2010 to generate emission inventories or emission rate lookup tables for on-road mobile sources (UNC-Chapel Hill and ENVIRON, 2014). We notice there are reports commenting that the NEI may overestimate the mobile NOx emission. For example, Anderson et al. (2014) estimated the NEI may overestimate mobile NOx emissions by 51–70%, based on the observed molar CO/NOx emission ratios from the DISCOVER-AQ campaign data. They argue that "the NEI overestimate of NOx emissions could indicate that engines produce less NOx and catalytic converters degrade more slowly than assumed by MOVES2010. MOVES2010 likely fails to capture dependence of NOx emissions on vehicle age accurately." We didn't explicitly explore the uncertainty of mobile NOx emission to the source apportionment results.

References:
UNC-Chapel Hill and ENVIRON International Corporation, Three-State Air Quality Modeling Study (3SAQS) – Final modeling protocol: 2011 emissions & air quality modeling platform,
http://vibe.cira.colostate.edu/wiki/Attachments/Modeling/3SAQS_2011_WRF_MPE_v8_draft_Aug04_2014.pdf
Anderson, D.C., Loughner, C.P., Diskin, G., Weinheimer, A., Canty, T.P., Salawitch, R.J., Worden, H.M., Fried, A., Mikoviny, T., Wisthaler, A. and Dickerson, R.R.,

Measured and modeled CO and NOy in DISCOVER-AQ: An evaluation of emissions and chemistry over the eastern US. Atmos. Environ., 96, 78-87, 2014.

*4.13 - 20: Does the inventory here contain the amount of NH3 from mobile sources mentioned in the introduction, or is if felt that this inventory under-represents this source?*

**Response:**

As mentioned in the previous response, the on-road mobile source is provided by MOVE2010, and it does account for the NH3 emissions from the mobile sources; see the attached picture below. However, these emissions are likely underestimated since recent work by Fenn et al., (2018), which was discussed in the manuscript, estimates that the 2011 NEI underestimates mobile NH3 emissions by a factor of 2.9.

[Figure]

Reference:

Fenn, M.E., Bytnerowicz, A., Schilling, S.L., Vallano, D.M., Zavaleta, E.S., Weiss, S.B., Morozumi, C., Geiser, L.H. and Hanks, K.: On-road emissions of ammonia: An underappreciated source of atmospheric nitrogen deposition, Sci. Total Environ., 625, 909-919, 2018.

*4.13 - 20: It would be very useful for answering these questions and others if the emissions totals by sector and species for the different tagged regions could be included in the supporting information and summarized in the text (as opposed to the summaries mentioned in the introduction, which reflect values in the literature but do not specifically refer to the values used in the modeling for this work).*

**Response:**

For this work, we used the 2011 NEI version 2 inventory from the EPA and updated the oil and gas sector at western U.S. based on the local survey data. As requested, we

provided the designated table (Table S2) in the supplemental material to provide the summary of 27 tagged regions in CAMx PSAT in this study and annual emissions for NH3 and NOx. The table is attached for reference.

Table S2. Summary of 27 tagged regions in CAMx PSAT in this study and their corresponding annual emissions for NH3 and NOx with agriculture (AG), oil and gas OG), wildfires and prescribed fires (fire), and remaining emission source sectors (Other). The items in the parentheses are aggregate regions based on prevailing wind patterns over the GYA for the source apportionment results reported in Figures 9–11.

| Tagged region | Total emission for nitrogen species (tons/yr) | | | | | | | | | |
| --- | --- | --- | --- | --- | --- | --- | --- | --- | --- | --- |
| | NH3 | | | | | NOx | | | | |
| | AG | OG | Fire | Other | total | AG | OG | Fire | Other | total |
| NW Colorado (Southwest) | 4,900 | 0 | 55 | 418 | 5,373 | 0 | 12,046 | 564 | 54,827 | 67,437 |
| NE Colorado (Southwest) | 37,041 | 0 | 415 | 3,157 | 40,613 | 0 | 16,002 | 749 | 72,830 | 89,581 |
| SE Colorado (Southwest) | 20,281 | 0 | 227 | 1,728 | 22,237 | 0 | 20,869 | 976 | 94,980 | 116,825 |
| SW Colorado (Southwest) | 6,672 | 0 | 75 | 569 | 7,315 | 0 | 5,504 | 258 | 25,051 | 30,812 |
| Upper Green River, Wyoming | 2,358 | 0 | 525 | 110 | 2,993 | 0 | 11,412 | 3,016 | 43,523 | 57,952 |
| Jackson, Wyoming | 2,375 | 0 | 529 | 111 | 3,015 | 0 | 477 | 126 | 1,817 | 2,420 |
| Eastern Wyoming (Other WY) | 7,298 | 0 | 1,625 | 342 | 9,265 | 0 | 3,013 | 796 | 11,490 | 15,299 |
| Western Wyoming (Other WY) | 18,046 | 0 | 4,018 | 845 | 22,910 | 0 | 10,925 | 2,887 | 41,662 | 55,474 |
| Yellowstone (Other WY) | 1,511 | 0 | 336 | 71 | 1,918 | 0 | 761 | 201 | 2,902 | 3,864 |
| Northern Idaho (Northwest) | 16,887 | 0 | 2,193 | 910 | 19,991 | 0 | 669 | 6,906 | 47,036 | 54,612 |
| Snake River Valley, Idaho | 43,696 | 0 | 5,674 | 2,356 | 51,726 | 0 | 682 | 7,030 | 47,882 | 55,594 |
| Northern Utah | 12,946 | 0 | 69 | 2,163 | 15,178 | 0 | 10,235 | 200 | 92,312 | 102,747 |
| Southern Utah (Southwest) | 10,083 | 0 | 54 | 1,685 | 11,822 | 0 | 8,907 | 174 | 80,338 | 89,419 |
| Nevada | 5,569 | 0 | 825 | 2,533 | 8,926 | 0 | 189 | 2,725 | 107,900 | 110,814 |
| Montana | 54,343 | 0 | 7,531 | 1,313 | 63,187 | 0 | 13,806 | 11,510 | 153,220 | 178,537 |
| Washington (Northwest) | 44,118 | 3 | 825 | 7,400 | 52,345 | 0 | 467 | 2,458 | 268,831 | 271,757 |
| Oregon (Northwest) | 43,626 | 0 | 8,858 | 5,164 | 57,649 | 0 | 925 | 28,231 | 146,062 | 175,218 |
| California | 203,204 | 155 | 3,056 | 111,240 | 317,655 | 0 | 8,806 | 9,457 | 669,421 | 687,684 |
| Mexico (Non U.S.) | | | | | 246,344 | | | | | 782,600 |
| New Mexico (Southwest) | 35,327 | 0 | 4,374 | 2,673 | 42,374 | 0 | 71,863 | 15,197 | 170,550 | 257,609 |
| Arizona (Southwest) | 33,247 | 0 | 9,041 | 8,520 | 50,808 | 0 | 1,489 | 26,817 | 250,201 | 278,506 |
| Texas&Oklahoma (Southwest) | 364,835 | 44 | 24,481 | 39,179 | 428,539 | 0 | 410,736 | 35,635 | 1,450,095 | 1,896,465 |
| Canada (Non U.S.) | | | | | 421,830 | | | | | 934,900 |
| North Dakota (Eastern U.S. + Great Plains) | 93,163 | 0 | 952 | 6,995 | 101,110 | 0 | 8,408 | 1,407 | 171,869 | 181,683 |
| Pacific (Non U.S.) | | | | | 292 | | | | | 251,698 |
| Far East U.S. (Eastern U.S. + Great Plains) | | | | | 2,627,200 | | | | | 9,296,000 |
| SD_KS_NE (Eastern U.S. + Great Plains) | 480,670 | 4 | 6,245 | 9,439 | 496,359 | 0 | 96,945 | 25,572 | 666,950 | 789,467 |
| Total: | | | | | 5,128,972 | | | | | 16,834,975 |

Also, we added a summary in the text about the emissions we used in this modeling study: "Table S2 provides the annual NH3 and NOx emissions used in this modeling study with a breakdown by tagged source regions and source sectors. Figure 2 provides the annual emissions of NH3 in the inner 12-km domain as well as the monitoring sites or receptor areas used for the model evaluation and analysis. For NH3 emissions, the AG sector contributed 84.1% of the total emissions within 12-km domain, while the OG, Fire, and Other sectors contributed 0.1%, 4.5%, and 11.4%, respectively (Table S2). In the Snake River valley, the AG sector emissions dominate the emission budget. For NOx emissions, the contribution rankings from the four tagged emission sources are Other (83.8%), OG (12.8%), Fire (3.2%), and AG (0%)."

*5.14: As anthropogenic SO2 emissions have declined in the US, the role of NOx and NH3 in forming ammonium nitrate aerosol has increased. How would PSAT account for the influence of the EGU sector via SO2 on deposition of PNH4 and PNO3, or is this not accounted for?*

**Response:**

We are not completely clear as to exactly what the reviewer is asking in this question. However, CAMx contains relatively complete chemical and thermodynamic mechanisms for inorganic sulfur and nitrogen gases and particles. Therefore, the interplay between $SO_2$ - $NO_x$ - $NH_3$ is accounted for in the model. For example, with the decreases in $SO_2$ emissions there should be more $NH_3$ available to neutralize $HNO_3$ forming particulate ammonium nitrate. The CAMx chemical and thermodynamic mechanism can account for these and other shifts and their impact on nitrogen deposition and be reflected in the PSAT source attribution results.

*6.9: Could the authors clarify what constituted questionable data, such that their results could be more reproducible?*

**Response:**

Questionable data refers to the measurements used to evaluate the model. There are certain protocols used by the measurement community to report their data and the associated credentials. For instance, for the wet deposition data reported by the NTN, a series of codes are assigned to samples that are considered invalid by the NTN for the purposes of computing weighted-mean concentrations, depositions, and data completeness estimates. The common reasons are contaminated samples, inadequate volume collected in the bucket for analysis, and lab error, for example. To make this statement clear, we changed the sentence from "All data flagged as questionable were removed from the analysis" to "All measurement data flagged as questionable, either due to maloperation or due to insufficient samples to calculate representative values, were excluded from the analysis. In Table 1, we also reported the percentage of validate measurements used for statistical analysis during evaluation time. For most of the nitrogen species, the percentage of validate samples are more than 80%."

We also added the percentages of measurement data completeness in the model performance evaluation table (Table 1) for reference.

*6.22: Does the mechanism for formation of N2O5 in CAMx match that in GEOS-Chem? If not, it's not clear how the reference to Heald et al. (2012) is relevant here.*

**Response:**

Thanks for pointing this out. The reference here is not proper. In GEOS-Chem, the inorganic chemistry mechanism used to model the pollutants' evolution from surface to

the stratopause is called the "tropchem" mechanism and is based on the NASA/JPL publication 10-6 for chemical kinetics and photochemical data for use in atmospheric studies. In total, 236 reactions were included in this mechanism, and reaction #225 has the parameterization of heterogeneous $N_2O_5$ reaction to form $HNO_3$ based on the ambient aerosol type, relative humidity, and temperature (Evans and Jacob, 2005). In CAMx, we used the CB6r2 mechanism, and it also includes consideration of this heterogeneous $HNO_3$ formation with the initial parameterization protocol as in Evans and Jacob (2005) but with revisions (Foley et al., 2010). However, since GEOS-Chem is a global photochemical model and the "tropchem" is different from a carbon bond mechanism, it is unfair to quote the evaluation statements regarding GEOS-Chem to the CAMx simulation results here. Therefore, we deleted this statement. Instead, we added two additional citations for reporting the same $HNO_3$ overestimation problem using regional air quality models (e.g., CMAQ, CAMx). Now this sentence read as:

"The overestimation of $HNO_3$ has also been reported in other regional-scale modeling simulations over the United States (e.g., Barker and Scheff. 2007, Foley et al., 2010; Thompson et al., 2015) with the carbon bond mechanism used in this study. The possible reason for the overestimation of $HNO_3$ may be due to the uncertainty for the $N_2O_5$ uptake coefficient setting for heterogeneous reactions (Foley et al., 2010)."

References:
    Baker, K. and Scheff, P.: Photochemical model performance for PM2. 5 sulfate, nitrate, ammonium, and precursor species SO2, HNO3, and NH3 at background monitor locations in the central and eastern United States, Atmos. Environ., 41, 6185-6195, 2007.
    Foley, K.M., Roselle, S.J., Appel, K.W., Bhave, P.V., Pleim, J.E., Otte, T.L., Mathur, R., Sarwar, G., Young, J.O., Gilliam, R.C. and Nolte, C.G., Incremental testing of the Community Multiscale Air Quality (CMAQ) modeling system version 4.7. Geosci. Model Dev., 3(1), 205-226, 2010.
    Evans, M.J. and Jacob, D.J., 2005. Impact of new laboratory studies of N2O5 hydrolysis on global model budgets of tropospheric nitrogen oxides, ozone, and OH. Geophysical Research Letters, 32(9).

*7.2: Is a unidirectional NH3 emission model expected to lead to larger NH3 concentrations in this region of the US than a bidirectional flux model?*
**Response:**
Currently, there is no bidirectional flux model for NH3 implemented in CAMx. The bidirectional flux model calculates the compensation point of NH3 between canopy and land-surface terrain and allows a portion of deposited NH3 to be emitted back into the atmosphere based on the emission potential of the soil NH3 pool. Conceptually, given the occurrence of re-emittance of certain amounts of NH3 into the atmosphere, the NH3

ground concentrations at the surrounding modeling grids (especially downwind grids) should be increased. The GYA area is adjacent and downwind of the Snake River valley and northern Utah, both of which have significant portions of agricultural sources (see Table S2). Therefore, it is a logical expectation that if the bidirectional NH3 model was implemented in CAMx, the bias in the simulated NH3 concentrations in this region would be decreased. Furthermore, in section 5, we discussed the potential benefit of including NH3 bidirectional parameterization into the CAMx model and the difficulties for implementation. To specifically address the reviewer's comment, we added the following statement:

"The poor NH3 results may be related to the high uncertainty in the NH3 emission inventory (Clarisse et al., 2009) and important missing physical mechanisms in the model, including the lack of bidirectional NH3 deposition (Zhang et al., 2010; Bash et al., 2013; Zhu et al., 2015). The GYA area is located downwind of the major agriculture sources in the Snake River valley and northern Utah (Table S2). The incorporation of the bidirectional NH3 flux mechanism in the model should increase ambient NH3 concentrations in the GYA and thus decrease the large model underestimation of NH3 concentrations."

*7.2: I would suspect that another possible factor leading to poor correlation and underestimation for NH3 is the overestimation of HNO3, which would promote excessive partitioning of NH3 to the particle phase. Did the authors consider evaluating NHx, or HNO3+PNO3, to get around the issues of partitioning (and thus hone in on issues related to sources and sinks)?*
**Response:**
It is possible that the poor model performance for NH3 may relate to the overestimation of HNO3 in the model, which would push excessive partitioning of NH3 into the particle phase. CAMx uses ISORROPIA to calculate the inorganic gas–particle thermodynamic equilibrium. From the old Table 1, we also see a slight overestimation of PNH4 in conjunction with the large underestimation of NH3 at CASTNET sites within the GYA. Therefore, we followed the suggestion of the reviewer to evaluate NHx to try to get around the possible bias in gas-particle partitioning. However, only a few locations existed within the GYA where a network has concurrent measurements of nitrogen gas and particulate species. We added the statistics for NH3, PNH4, and NHx model performance during the GrandTReNDS campaign at the three sites in the updated Table 1 (attached below).

Table 1. CAMx model performance for nitrogen species concentrations as well as nitrogen dry/wet depositions evaluated at sites in AMoN, CASTNet, IMPROVE, NTN

networks as well as the 3 sites during GrandTRENDS campaign over the GYA region (see Figure 1 for site locations) in 2011.

| Species | | Network | Duration | OBS[a] | SIM[b] | #Site[c] | N[d] (% completeness) | R[e] | NMB[f] | NME[g] | FB[h] | FE[i] |
|---|---|---|---|---|---|---|---|---|---|---|---|---|
| concentration | $NH_3$ (ppb) | AMoN[1] | Sep 22-Dec 12 | 0.49 | 0.30 | 1 | 7 (100%) | 0.20 | -65% | 67% | -52% | 53% |
| | | GrandTReNDS[2] | Apr 5-Sep 21 | 0.55 | 0.46 | 3 | 434(97.7%) | 0.30 | -16% | 57% | -42% | 63% |
| | $HNO_3$ (ppb) | CASTNet[3] | Jan 4-Dec 27 | 0.23 | 0.47 | 2 | 83(98.8%) | 0.72 | 108% | 117% | 60% | 71% |
| | | GrandTReNDS[2] | Apr 5-Sep 21 | 0.28 | 0.54 | 3 | 435(97.9%) | 0.60 | 106% | 109% | 63% | 68% |
| | $PNO_3$ ($\mu g\ m^{-3}$) | CASTNet[3] | Jan 4-Dec 27 | 0.19 | 0.25 | 2 | 83(98.8%) | 0.42 | 37% | 76% | 26% | 64% |
| | | IMPROVE[4] | Jan 3-Dec 29 | 0.14 | 0.22 | 4 | 332(68.5%) | 0.35 | 58% | 108% | 51% | 80% |
| | | GrandTReNDS[2] | Apr 5-Sep 21 | 0.13 | 0.15 | 3 | 435(97.9%) | 0.45 | 15% | 71% | 14% | 60% |
| | $PNH_4$ ($\mu g\ m^{-3}$) | CASTNet[3] | Jan 4-Dec 27 | 0.17 | 0.18 | 2 | 83(98.8%) | 0.28 | 3% | 39% | 7% | 41% |
| | | GrandTReNDS[2] | Apr 5-Sep 21 | 0.14 | 0.17 | 3 | 433(97.7%) | 0.12 | 23% | 64% | 34% | 61% |
| | $NH_x$ ($\mu g\ m^{-3}$)[4] | GrandTReNDS[2] | Apr 5-Sep 21 | 0.68 | 0.63 | 3 | 427(96.2%) | 0.26 | -7% | 48% | -22% | 46% |
| N Deposition | $HNO_3$ dry (kg N ha[-1]) | CASTNet[3] | Jan 4-Dec 27 | 0.071 | 0.187 | 2 | 83(98.8%) | 0.81 | 153% | 156% | 77% | 82% |
| | | GrandTReNDS[2] | Apr 5-Sep 21 | 0.016 | 0.049 | 3 | 435(97.9%) | 0.66 | 204% | 209% | 101% | 104% |
| | $PNO3$ dry (kg N ha[-1]) | CASTNet[3] | Jan 4-Dec 27 | 0.012 | 0.023 | 2 | 83(98.8%) | 0.14 | 96% | 148% | 48% | 97% |
| | | GrandTReNDS[2] | Apr 5-Sep 21 | 0.010 | 0.011 | 3 | 435(97.9%) | 0.61 | 8% | 58% | 1% | 65% |
| | $PNH_4$ dry (kg N ha[-1]) | CASTNet[3] | Jan 4-Dec 27 | 0.018 | 0.019 | 2 | 83(98.8%) | 0.1 | 7% | 57% | 22% | 61% |
| | | GrandTReNDS[2] | Apr 5-Sep 21 | 0.006 | 0.004 | 3 | 433(97.7%) | 0.1 | -33% | 46% | -28% | 53% |
| | $NO_3^-$ wet (kg N ha[-1]) | NTN[5] | Jan 4-Dec 27 | 0.079 | 0.097 | 5 | 214(82.3%) | 0.34 | 31% | 126% | 12% | 100% |
| | | GrandTReNDS[2] | Apr 5-Sep 21 | 0.051 | 0.083 | 3 | 427(96.2%) | 0.15 | 60% | 94% | 42% | 71% |
| | $NH_4^+$ wet (kg N ha[-1]) | NTN[5] | Jan 4-Dec 27 | 0.088 | 0.126 | 5 | 214(82.3%) | 0.32 | 49% | 142% | 19% | 106% |
| | | GrandTReNDS[2] | Apr 5-Sep 21 | 0.103 | 0.147 | 3 | 427(96.2%) | 0.48 | 42% | 72% | 30% | 64% |
| | Precipitation (cm) | NTN[5] | Jan 4-Dec 27 | 0.77 | 2.34 | 5 | 214(82.3%) | 0.54 | 215% | 242% | 64% | 118% |
| | | GrandTReNDS[2] | Apr 5-Sep 21 | 0.33 | 0.95 | 3 | 427(96.2%) | 0.42 | 187% | 207% | 69% | 94% |

The time series plots with the daily mean concentration comparisons are also given below. The CAMx model still underestimates the NH3 concentration (NMB = -16%) and overestimates PNH4 concentration (NMB =23%) at the three sites, but if we evaluate NHx, the model bias is smaller (NMB = -7%).

[Figure]

Also, we added a sentence in the first paragraph of section 3.2 as:
"The underestimation of NH3 concentration still existed (NMB = -16%), and one of the possible reasons may be due to the overestimation of HNO3 in the model pushing excessive partitioning of NH3 into the particle phase, which can be shown by the better model performance for NHx simulation (NMB = -7%) without splitting the gas-particle partition bias."

*7.7: Are the performance metrics referenced here relevant for a study focusing on Nr source attribution? I could imagine if a studies goal was to forecast total PM2.5 concentrations, then opposing large biases in e.g. NH3 vs HNO3 would be of little concern; here, these issues seem much more considerable in terms of their impact on the final conclusions. Overall, I think the authors need to do more work in this regards to convince the readers of the merits of the application of the model so SA in the presence of such errors and biases.*

**Response:**

The performance metrics referenced here from Simon et al. (2012) are the compilation of 69 peer-reviewed articles published between 2006 and 2012 focusing on regional air quality model performance evaluation for total PM2.5, speciated PM2.5, and wet deposition of sulfate, nitrate, and ammonium over the United States and Canada. None of the simulations compiled by the authors focus on the $N_r$ source attribution. Reviewer #1 also has suggestions on this sentence. In here we just want to demonstrate that our CAMx base case modeling performance is in line with the peer modeling results and provides a good platform for further source attribution analysis. We provided Table S3 in the supplemental material to summarize the collected recent model performance evaluations for nitrogen species and revised this sentence to:

"Table S3 provides a comparison of regional CTM performance evaluations against measured N- containing species over the United States from peer-reviewed studies in recent years (e.g., Simon et al., 2012; Bash et al., 2013; Zhang et al., 2013; Yu et al., 2014; Thompson et al., 2015; Li et al., 2017). The model performance results in this study are comparable to these past studies including the overestimation of HNO3 and underestimation of NH3. Resolution of these biases requires additional research and these biases need to be taken into account when interpreting the source attribution of Nr deposition within the GYA."

*Fig 3: I find it interesting that the measurements at each site show a distinct reduction in NH3 dry dep in September, whereas CAMx shows a maximum in September for Driggs and Grand Targhee. Can authors comment on this?*

**Response:**

The monthly dry NH3 deposition values at the three sites associated with Figure 3 (now Figure 4) are attached below as a Table for clarification. It is true that the NH3 dry deposition (light blue in the figure) in September at each site shows a distinct reduction compared with the previous month (0.094 versus 0.209 in Driggs, 0.074 versus 0.147 in Grand Targhee, and 0.049 versus 0.113 in NOAA), but the corresponding CAMx results have the opposite trend for the Driggs and Grand Targhee sites.

|  |  | GrandTReNDS | CAMx |
|---|---|---|---|
|  |  | (kg N/ha) | (kg N/ha) |
| Driggs | Apr | 0.114 | 0.142 |
|  | May | 0.158 | 0.104 |
|  | Jun | 0.156 | 0.104 |
|  | Jul | 0.194 | 0.101 |
|  | Aug | 0.209 | 0.134 |
|  | Sep | 0.094 | 0.194 |
| Grand Targhee | Jul | 0.018 | 0.071 |
|  | Aug | 0.147 | 0.101 |
|  | Sep | 0.074 | 0.119 |
| NOAA CC | May | 0.018 | 0.043 |
|  | Jun | 0.076 | 0.050 |
|  | Jul | 0.085 | 0.049 |
|  | Aug | 0.113 | 0.102 |
|  | Sep | 0.049 | 0.088 |

Back trajectory analysis shows that during the GrandTReNDS campaign period, the dominant source origins impacting the Nr in the GYA are from Snake River valley and northern Utah (Prenni et al., 2015). The high NH3 deposition at the three sites in September in the CAMx simulation results is also verified with the spatial plots attached below. The high deposition is associated with the high NH3 emission rates in September from the Snake River valley.

[Figure]

[Figure]

More importantly, if we compare the monthly mean dry deposition velocities used to calculate the measured NH3 dry deposition with the corresponding CAMx values, we find that there is a steep jump from August to September from the GrandTReNDS calculations, while the deposition velocity values from the models keep steady. Therefore, we believe this discrepancy is mainly due to the different variation trend of dry deposition velocity between the measurements and the model.

[Figure]

We revised the corresponding sentences in section 3.2 as:

"As shown, the simulation does a poor job of reproducing the total Nr deposition rates both in the month-to-month variation as well as across the sites. The difference in the dry NH3 deposition monthly variation between measurements and simulation is mainly due to the difference in associated dry deposition velocity used for calculation. However, consistent with the observations, the simulation shows that wet deposition is larger than dry and that the contribution from reduced N deposition was larger than from the

oxidized N deposition at all three sites, although the observed range of 70–80% reduced N was more than the 55–68% simulated in CAMx."

Reference:

    Prenni, A.J., Levin, E.J.T., Benedict, K.B., Sullivan, A.P., Schurman, M.I., Gebhart, K.A., Day, D.E., Carrico, C.M., Malm, W.C., Schichtel, B.A. and Collett, J.L., Gas-phase reactive nitrogen near Grand Teton National Park: Impacts of transport, anthropogenic emissions, and biomass burning. Atmos. Environ., 89, 749-756, 2014.

*12: How much did reducing the NH3 dry deposition change the total NH3 deposition amounts and their underestimation compared to observations mentioned in previous sections?*

**Response:**

In the supplemental material, Figure S6, we updated the change of spatial patterns of the simulated total NH3 deposition over the GYA during July–August 2011 due to the change of NH3 deposition velocity in CAMx (the middle panel in the attached figure below).

[Figure]

Figure S6. Change of spatial patterns of the simulated total Nr deposition (top panel), total NH3 deposition (middle panel) as well as contributions from agricultural emissions

sector to total Nr deposition budget (bottom panel) over the Greater Yellowstone Area (GYA) during July–August 2011 due to the change of NH3 deposition velocity in CAMx.

Attached table shows the dry and wet nitrogen deposition change at the GYA due to changing NH3 deposition velocity in CAMx during July-August 2011. Decreasing the NH3 deposition velocity will increase the NH3 surface concentration and improve the model bias for underestimation (see Figure S5). Still, the total NH3 dry deposition in the GYA will decrease by 3%. However, the NH3 wet deposition in the GYA is significantly increased (73%) due to longer NH3 lifetime since emit and further deposition into the GYA during precipitation events. On average, a 31% increase for total Nr deposition from the agriculture source sector (which is dominated by NH3 emissions) can be seen by decreasing the NH3 dry deposition velocity.

| | base (kg N/ha) | | | DV_0.1 (kg N/ha) | | | difference(%) | | |
|---|---|---|---|---|---|---|---|---|---|
| | Dry | Wet | Total | Dry | Wet | Total | Dry | Wet | Total |
| BC | 0.033 | 0.040 | 0.073 | 0.029 | 0.045 | 0.074 | -12.5% | 14.3% | 2.1% |
| Agriculture | 0.038 | 0.030 | 0.069 | 0.037 | 0.052 | 0.090 | -2.7% | 73.1% | 30.8% |
| Oil&Gas | 0.004 | 0.001 | 0.005 | 0.004 | 0.002 | 0.005 | -1.2% | 14.1% | 3.2% |
| Other+Fire | 0.149 | 0.056 | 0.206 | 0.130 | 0.070 | 0.200 | -13.2% | 25.6% | -2.7% |
| Total | 0.224 | 0.128 | 0.352 | 0.199 | 0.170 | 0.369 | -11.2% | 33.2% | 4.9% |

*13.1: It seems like earlier there were several possible reasons for this, such as overestimated HNO3 concentrations, and yet here only precipitation biases are considered?*

**Response:**

Due to the limited amount of computational resources, we didn't conduct the HNO3 sensitivity study or quantify its impact to source apportionment results. It is true that the overestimation of HNO3 concentration is a major uncertainty for the simulated nitrogen deposition budgets (see Figure 3 and Figure 4). Heald et al. (2012) used GOES-Chem to simulate inorganic aerosol loading and NH3 concentrations over the United States. They also reported significant overestimation of HNO3 concentrations and found that by reducing HNO3 concentrations to 75% of their simulated values, the model can correct the bias in nitrate as well as in ammonium simulation. They didn't pinpoint the mechanism underneath this model performance improvement but provided a general statement that it may be due to "a combination of errors in chemistry, deposition and sub-grid near-surface gradients." However, the findings from Heald et al. (2012) using GEOS-Chem are hard to refer here to justify the similar impact from CAMx given the differences of those two photochemical models in terms of implementation scales (regional versus global) and chemical mechanism (carbon bond versus tropchem). We expect the decrease of deposition of oxidized nitrogen in the GYA by decreasing the HNO3 concentrations in the model and we suspect the impact from further source regions with high NOx emissions will become smaller to the GYA.

We added a sentence at the section 5 as:

"The overestimation of HNO3 concentrations in the GYA is another reason for the wet Nr deposition overestimation. However, its impact on source apportionment results was not conducted here due to unclear reasons for the model bias (emission, chemistry, meteorology, deposition scheme) and limited computational resources."

Reference:

Heald, C.L., Collett Jr, J.L., Lee, T., Benedict, K.B., Schwandner, F.M., Li, Y., Clarisse, L., Hurtmans, D.R., Van Damme, M., Clerbaux, C. and Coheur, P.F., Atmospheric ammonia and particulate inorganic nitrogen over the United States. Atmos. Chem. Phys., 12(21), 10295-10312, doi:10.5194/acp-12-10295-2012, 2012.